# Team Size and Composition in Home Healthcare: Quantitative Insights and Six Model-Based Principles

**DOI:** 10.3390/healthcare11222935

**Published:** 2023-11-09

**Authors:** Yoram Clapper, Witek ten Hove, René Bekker, Dennis Moeke

**Affiliations:** 1Department of Mathematics, Vrije Universiteit Amsterdam, 1081 HV Amsterdam, The Netherlands; y.clapper@vu.nl (Y.C.); r.bekker@vu.nl (R.B.); 2Academy of Organization and Development, HAN University of Applied Sciences, 6826 CC Arnhem, The Netherlands; witek.tenhove@han.nl

**Keywords:** home care, work force, resource allocation, efficiency

## Abstract

The aim of this constructive study was to develop model-based principles to provide guidance to managers and policy makers when making decisions about team size and composition in the context of home healthcare. Six model-based principles were developed based on extensive data analysis and in close interaction with practice. In particular, the principles involve insights in capacity planning, travel time, available effective capacity, contract types, and team manageability. The principles are formalized in terms of elementary mathematical models that capture the essence of decision-making. Numerical results based on real-life scenarios reveal that efficiency improves with team size, albeit more prominently for smaller teams due to diminishing returns. Moreover, it is demonstrated that the complexity of managing and coordinating a team becomes increasingly more difficult as team size grows. An estimate for travel time is provided given the size and territory of a team, as well as an upper bound for the fraction of full-time contracts, if split shifts are to be avoided. Overall, it can be concluded that an ideally sized team should serve (at least) around a few hundreds care hours per week.

## 1. Introduction

As in many other Western European countries, the long-term sustainability of Dutch home healthcare (HHC) system (here, we define home healthcare as an array of health and social support services provided to clients in their own residence [1]) is under serious pressure. On the one hand, we see an increase in demand due to an aging population, and a shift from care provided in an institutional setting to providing care closer to the care user’s own home environment. By 2040, it is projected that one in four Dutch people will be aged 65 or over, and the number of persons older than 80 years is expected to almost double from 0.9 million in 2021 to over 1.6 million in 2040 [2]. This will raise the pressure on the Dutch HHC system because the prevalence of physical or mental disability increases with age [3]. When it comes to the shift of institutional care towards the home environment, it is mainly triggered by two developments: extramuralisation and medical care at home. In this context, extramuralisation can be described as government policy that aims to shift from providing care in a nursing home setting to care at home. Medical care at home is an alternative to (more expensive) inpatient hospital admission, enabling patients to receive hospital-level care at or closer to home.

On the other hand, looking at the supply side we see that the availability of healthcare professionals is under increasing pressure due to labor market tightness and high absenteeism rates. The Dutch healthcare and welfare forecast model shows that the shortage of healthcare workers will increase from 55,000 people in 2023 to about 155,000 people in 2032, with the largest shortages expected in nursing home and home healthcare [4]. Due to the increasing staff shortages, the healthcare sector risks falling into a vicious circle: staff shortages create an increased workload, which leads to more absenteeism, thereby creating an even greater shortage, etc. For example, that this vicious circle is lurking becomes evident from the increasing absenteeism rates. In 2022, the average absenteeism rate in the healthcare sector reached about 8.3% [5]. This is the highest absence rate ever measured and represents an increase of 15% compared to the year 2021.

Because of these challenges, many Dutch HHC providers are searching for ways to improve the balance between the available workforce capacity (i.e., supply) and the needs and preferences of their clients (i.e., demand). In collaboration with our partner HHC organization, we encountered (at least) two fundamental problems in obtaining the appropriate balance. The first problem is the lack of insight into the current demand. Although demand prospects exist for both individual clients, as well as some occasional handcrafted estimates at a more aggregate level, structural monitoring of the total demand requirements of the complete client base is yet uncommon in HHC. The second problem relates to the appropriate dimensioning of teams. In fact, the elementary initial question of our partner HHC organization was ‘What is the ideal team size?’. Obviously, this question can be addressed from multiple angles; our aim is to provide some generic rules of thumb for determining the scale at which teams should be organized. In other words, our key research question is as follows: To what extent is it beneficial to utilize the potential of economies of scale in HHC?

### 1.1. Background: Existing Literature

The problems faced by HHC providers have triggered a body of research over the past decade. In the relatively early study of Matta et al., a framework to model HHC organization from an operations management perspective is proposed, whereas a taxonomic classification is provided by Hulshof et al. [6,7]. From those papers, it is evident that the number of studies focusing on strategic and/or tactical decisions in HHC, from an operations management (OM) and operations research (OR) perspective, is very limited. This conclusion is confirmed by the overview paper of Grieco et al. [8]. According to Grieco et al., the vast majority of the OR-related HHC studies focus on staff-to-patient allocation, visit scheduling, and the routing of visits, leading to more technical reviews concerning HHC routing and scheduling [9,10,11]. In contrast, only a few studies consider strategic and/or tactical decisions concerning team size and composition.

Below, we provide an overview of and discuss relevant studies that focus on resource dimensioning and team composition within a long-term care setting (i.e., residential and HHC). We opted for this scope because HHC is a prominent part of the Dutch long-term care (LTC) system. In addition, as resource dimensioning and team composition in an LTC setting is generally carried out at strategic and tactical levels, studies that focus on the operational level were not taken into account. In addition to resource dimensioning and team composition, the concept of ’economies of scale’ will also be elaborated on as it plays an important role in the remainder of the paper.

From a healthcare-capacity-planning perspective, determining team sizes can be considered a resource dimensioning issue. Many studies have been devoted to resource dimensioning, most frequently for hospital capacity, of which a large share are quantitative in nature and stem from the domains of OR and OM.

Resource dimensioning at a strategic level involves structural decision-making on a relatively long time horizon (typically 1 year or more). Two examples of OR/OM studies involving residential care services at a strategic level are those of Christensen et al. and Moeke et al. [12,13]. Multiple quantitative studies have been conducted regarding economies of scale and scope from the perspective of the total organization (i.e., applying an aggregate perspective). Most of these studies make use of methods like regression analysis (e.g., [12]) or data envelopment analysis (e.g., [14]). Using a less aggregate approach, the study of red Moeke et al. aims to provide more insight into the effects of scale for small-scale living facilities in terms of waiting time and occupancy [13]. They present a comprehensive what-if analysis based on a discrete-event simulation model. When it comes to OR/OM literature regarding resource dimensioning on a strategic level in a home healthcare context, the districting problem is the most common area of focus (see, e.g., [15,16]). According to Benzarti et al., *districting a territory is a strategic decision that aims at grouping basic units (a set of patients) into larger clusters, i.e., districts, so that these districts are “good” according to relevant criteria. These criteria can be related to the activity, demography, or geographic characteristics of the basic units* [15]. In this paper, we essentially also try to determine the ‘optimal size of a district’, but our goal is to provide generic guidelines that abstract from the specific region. As such, the study presented in the current paper has a fundamentally different focus than the districting papers mentioned above.

The time horizon of decisions related to resource dimensioning at a tactical level is typically 3–12 months. OR/OM studies with a focus on resource dimensioning at a tactical level in a residential care context are scarce. The studies of Moeke et al. and Van Eeden et al. were the only ones we could find [17,18]. The study of Moeke et al. provides insights into how and why ‘scale of scheduling’ and the enlargement of care workers’ jobs (blending tasks of different qualification levels) affect the number and type of staff required to meet the preferences (in terms of day and time) of nursing home residents [17]. The focus is on activities of daily living (i.e., activities like bathing or showering, dressing, getting in and out of bed or a chair, walking, using the toilet, and eating). The study of Van Eeden et al., on the other hand, focuses on determining the required amount of capacity regarding random care activities [18]. Based on the analysis of real-life ‘call button’ data, they present a queueing model that can be used by nursing home managers to determine the number of care workers required to meet a specific service level. As mentioned in the recent overview of Grieco et al., OR/OM studies with a focus on resource dimensioning at a tactical level are also scarce in a home healthcare context [8]. To the best of our knowledge, only the following three studies fall into this category: [19,20,21]. Each of the three studies propose a two-stage capacity planning approach based on (integer linear) stochastic programming. The model of Nikzad et al. considers decisions on districting, staff dimensioning, resource assignment, scheduling, and routing simultaneously [19]. Their results show that the algorithm is able to solve large instances. As such, it also considers the more strategic issue of districting. In the work of Restrepo et al., a two-stage stochastic programming model is presented for employee staffing and scheduling in a HHC context [20]. In this model, the issue of staff dimensioning is part of the first-stage decision process. Finally, Rodriguez et al. aim to determine the number of care workers required to balance the coverage of patients in a region and the workforce cost over several months [21]. We observe that these papers tend to focus on optimization rather than on providing generic capacity guidelines, which is our aim.

Team composition also plays an important role in creating effective and efficient LTC delivery systems (see e.g., [22]). In line with the objective of this study, our literature search focused on OR/OM studies that deal with determining the optimal skill-mix in an LTC context, where skill-mix refers to *the mix of staff in the workforce or the demarcation of roles and activities among different categories of staff* [23]. More specifically, regarding team composition, the focus in this study is on ‘the mix of staff’ in terms of qualification levels (see also Section 3). Despite determining the right staff mix being considered important, to the best of our knowledge, the work of Moeke et al. is the only OR/OM study that focuses on this issue in an LTC context [17].

Finally, we note that questions concerning resource dimensioning and ‘team size’ are intimately linked to the concept of economies of scale. For instance, for bed capacity decisions in hospitals (phrased as ‘how many hospital beds’ by Green [24]) it has long been recognized that smaller hospital units should have lower target occupancy rates to achieve the same levels of delay. More generally, economies of scale in resource planning describes the positive relationship between the performance of the planning outcome (in terms of efficiency or effectiveness) and the pooling of customer demands, along with the pooling of the required resources to serve those demands. Therefore, within the realm of resource planning, it also known as the ’pooling principle’ [25,26,27]. From a mathematical perspective, the benefits of increasing scale are the consequence of a reduction in relative variability as the standard deviation of the sum of two random variables is smaller than the sum of the two standard deviations (if the coefficient of correlation is smaller than 1) [28]. We refer to the studies of Van Leeuwaarden and Whitt for a more elaborate exposition of the impact of scale in a queueing context [29,30].

### 1.2. Background: Practice

In the Netherlands, HHC services are provided to persons in need of care or support due to (chronic) illness, disability, or impairment. Determining eligibility for HHC services is carried out by the Centre for Healthcare Indication (CIZ). To receive paid home care, the CIZ must issue a so-called ‘indication of need’. The services provided by HHC organizations must fit within the limitations of this indication of need (the type of care, amount of care, time period, etc.). In 2021, the Netherlands counted over 2500 HHC providers who collectively served about 585,000 people in the same year [31]. Especially, elderly people make use of HHC services. By 2021, roughly about 80% of the Dutch HHC clients were aged 66 and over, with an average age of 75. Most recipients are women (59%), which can be explained by the fact that life expectancy is higher for women [32].

Our partner HHC provider provides home, residential, custodial, personal, and informal care support to roughly 12,000 clients, with about 4000 employees (1600 FTE) and 1000 volunteers (2021 data). Regular home care is provided by a group of 55 teams. During the period 2020–2021, each team served 210 clients on average. The variation in client numbers across teams largely hinges on the intensity of care required by the clients and the population density of the relevant area. The total coverage area of the HHC teams is around 300 square kilometers. Table 1 provides an overview of the main characteristics per type of area (i.e., urban, suburban, and rural).

Within each team, not every care worker is allowed to perform all tasks. Based on their education and expertise, care workers are hierarchically divided into three distinct qualification levels (QLs). Depending on the type of care, healthcare tasks are assigned to a healthcare worker with the required level of qualification. The hierarchical division of care workers’ tasks is also referred to as differentiated practice (e.g., [17]). Table 2 shows the QLs relevant in the context of this study. Here, the three QLs are denoted as PV niveau 2+, PV niveau 3, and VP niveau 3, as described by our partner HHC organization. Note that ‘PV’ and ‘VP’ are Dutch abbreviations for ‘persoonlijke verzorging’ (personal care) and ‘verpleegkundigde zorg’ (nursing), respectively, whereas the number after ‘niveau’ (level) denotes the required skill level.

In the context of capacity planning, non-direct care activities should be taken into account for the working time of the care workers (e.g., [28]). Regarding the division of working time, we use the classification as presented in Figure 1. The percentages that correspond to the categories ‘non-client related administration time’, ‘holiday and leave time’, and ‘sick leave’ are according to the studies [5,33,34], respectively.

The sum of the first two categories is the time during which care workers are available; we will refer to this as the *effective capacity* (as opposed to categories three and four in which care workers are not available). In the remainder of this paper, we mainly consider the effective capacity, with a particular focus on direct care time (e.g., Section 3.1 below also only involves direct care). The holiday and (sick) leave time are assumed to be given.

### 1.3. Contribution

The blind spot in the existing literature, enhanced by the demand of our partner organization and the current challenges faced by Dutch HHC organizations, has been the primary motivation for a constructive research study whose results are presented and discussed in this paper. The aim of this paper is to provide guidance to managers and policy makers by formulating a set of six practically applicable principles that address issues on capacity planning regarding team size and composition in a HHC context. The team size concerns the required number of care workers per team, whereas the team composition refers to the mix of care workers (in terms of qualification levels) in each team and the demarcation of roles and activities among the different categories of care workers. To address the issue of ‘the ideal team size’, we do not necessarily restrict ourselves to the current division of teams; the goal is to provide insight into the impact of changing the amount of demand that should be served by a single team. This can be practically achieved by either (re-)designing the cooperation between teams or, more drastically, by splitting or merging teams or redesigning the current division (which relates to the districting problem; see, e.g., [8]).

The six principles originate from discussions with our partner HHC organization and are formalized in terms of elementary mathematical models that capture the fundamental elements. Furthermore, the principles are supported by real-life data and demonstrated using practice-based scenarios. The paper is of value to both the management of HHC organizations as well as the scientific research community. For management, the principles provide both guidance and quantitative support regarding decisions about the employment of capacity, including strategic questions concerning the ‘ideal team size’. A particularly appealing property of the presented principles is that they support decision-making without the need for detailed data. For the scientific community, the data analysis offers insight into some key characteristics of HHC. Moreover, the models considered in this paper are of a fundamental nature; they may serve as inspiration and a starting point for more detailed modeling of the HHC demand and supply processes.

## 2. Methods

In accordance with the approach for constructive research presented by Kasanen et al., the following steps were followed [35]. With help of our partner HHC organization, we first identified a practical problem with research potential. Next, to gain a more comprehensive understanding of the topic, we conducted an extensive and systematic data analysis (see Section 3.1). To this end, we obtained data from our partner organization regarding all planned care activities of the years 2020 and 2021. For each single activity, we have, among others, an anonymized client ID, the date and day part, the duration, the qualification level, and the location (estimate). We note that we were unable to obtain historical data about the deployment and contracts of care workers (the capacity of the service system). As such, the number of care workers is based on generic estimations. As part of the data-validation process and the corresponding analysis, regular monthly validation sessions were conducted throughout the project. These sessions involved collaboration with professionals from our partner HHC organization. Guided by the data analysis outlined in Section 3.1 and in close interaction with our partner organization, we then developed six model-based principles (i.e., rules of thumb) (see Section 3.2). For these six principles, we formulated elementary mathematical models that capture the essential properties of each principle. Subsequently, using various forms of algebraic manipulations, we obtained the performance measures of interest for each model. Next, by using practice-based scenarios, we demonstrated the added value from a practical perspective (see Section 4). Finally, we discuss the applicability and scientific value of the presented principles (see Section 5).

## 3. Results

In Section 3.1, we present the results of our data analysis. Furthermore, the model-based principles are presented and elaborated on in Section 3.2.

### 3.1. Data Analysis

The aim of this subsection is to provide insight into the demand for home care, where the demand is defined as the planned HHC activities over time. Although the delivery of care is influenced by how capacity is deployed, we use the planned care activities as an approximation of the actual demand. For interpretation, it is useful to consider a period during which a client regularly receives the same type of care, which we refer to as a *case*. More specifically, a case is defined as care for one client at one given qualification level, for which the time difference between two subsequent visits does not exceed 30 days. In practice, one client may have multiple active cases simultaneously. We first consider the total demand for care (volume of care) revealing substantial variability in demand. Subsequently, we decompose the volume of care into its three primary ingredients: demand per case, the number of new cases per week, and the length of stay (LoS) per case.

#### 3.1.1. Volume of Care

For the considered qualification levels, with a total of about 76%, the vast majority of the delivered care consists of personal care, i.e., PV niveau 2+ and PV niveau 3 (see Table 2 for further details). Personal care encompasses all actions and practices that individuals typically undertake to maintain their well-being. This includes not only basic personal hygiene routines, such as bathing, but also specialized personal care required to address health conditions, such as managing a stoma.

The distribution of care provided among teams, QLs, and area types is depicted in Figure A1, showing the aggregate demand over the years 2020 and 2021. In line with earlier observations, the distribution of care types over the three QLs remains predominantly occupied by personal care. Furthermore, no significant differences can be observed between the various area types. The average aggregate planned care per team is 228 h per week. For most teams, the demand is reasonably close to this average, albeit there are some smaller (teams 5, 27, and 28 have a total of less than 15k care hours), and larger (teams 4, 21, 23, 31, 32, 37, and 41 have more than 30k care hours) teams. In Figure 2, boxplots of the total weekly demand per team are depicted. As indicated by both the interquartile range and the differences between the upper and lower whiskers, there is considerable variability in the aggregate demand per team. In the figure, large volumes of weekly demand are typically associated with more variability, but this does clearly not apply to all teams. The weekly demand per team is more or less symmetric, with only a few teams exhibiting stronger degrees of skewness (left and right). Observe that the variability in weekly demand makes the efficient use of capacity challenging, as we will demonstrate in Section 3.2.1 and Section 3.2.4.

The total demand for care per weekday and for each part of the day is visualized in Figure 3. The demand over the course of the week is rather stable, with a decrease in demand during the weekend of about 19.3% compared to the weekdays. The differences in demand across the day are more noticeable. The vast majority of care is taking place in the morning (68.6%), followed by the evening (24.2%). Only 7.3% of the demand is provided during the afternoon. This uneven distribution of demand across the day may complicate the deployment of care workers, as we will demonstrate in Section 3.2.5.

#### 3.1.2. Case Demand

The boxplot in Figure A2a describes the distribution of mean weekly care per case over all 55 teams. The median is about 3.3 h, with the lower and upper fences at 2.4 and 4.0 h, respectively. To put this into perspective, in the year of 2021, HHC clients in the Netherlands received an average of 6 h of care per week. However, the variation in the received amount of care was large. For example, terminally ill clients received 25 h of care per week, while frail elderly and chronically ill people who were in need of somatic and/or psycho-geriatric care for more than 3 months received 4 h of support per week [32]. Note that a client may have multiple cases simultaneously, making a comparison between care per case and care per client more difficult.

To visualize the variability in weekly planned care per case, Figure A2b depicts a boxplot of the variance-to-mean ratios (VMRs) of the weekly care per case of the 55 teams. The figure indicates that the variance in demand per case is roughly about four times the mean (50% of the values are between 3.1 and 4.5).

#### 3.1.3. Arrival Rate

The boxplot in Figure A2c illustrates the distribution of mean weekly new case arrivals across the 55 HHC teams. The median number of mean weekly new case arrivals by team equals 3.6, with lower and upper fences at 1.0 and 6.5, respectively. The median of the 55 VMRs of the number of new cases is 1.4, and the VMRs are somewhat right skewed with a lower fence at 0.9 and an upper fence at 2.4 (Figure A2d). Note that there is thus some slight overdispersion compared to a Poisson arrival process.

#### 3.1.4. Length of Stay

The LoS is here defined as the number of weeks between the start and conclusion of a given case. To estimate the distribution of the LoS and corresponding statistical measures, a Kaplan–Meier (KM) survival curve $\hat{S}\left(t\right)$ is constructed. This is the solid line in Figure A3. The curve of the tail distribution stops around the 100 week mark, which is the actual time range of the data set. This highlights a key challenge when estimating the mean and variance of the LoS as the data are both left- and right-censored.

A common approach to handling censored data is to fit a parametric distribution, such as the Weibull, to the non-parametric KM curve. Figure A3 depicts the Weibull fit to the KM curve for one care team. Although the quality of the fit seems to be (visually) acceptable for this particular case, an accurate estimation of the tail seems difficult, which in turn may severely impact the estimation of the mean and variance. For the analysis in Section 3.2, we therefore use an implied mean LoS.

### 3.2. Model-Based Principles

In this section, we formulate six key principles that can be used as a guideline for tactical and strategic decisions regarding the team size and composition. The principles are based on stylized models that capture the essential dynamics of the HHC process. For each principle, a more complex model can be constructed. However, our focus here is on simplicity, and we aim to facilitate a shift in the mindset of the HHC managers.

Below, we first state the principles in popular terms. More precise statements are presented in the subsequent Section 3.2.1, Section 3.2.2, Section 3.2.3, Section 3.2.4, Section 3.2.5 and Section 3.2.6. In each subsection, we first present the model on which the principle is based, followed by a numerical illustration.

**Principle** **1**(Care demand)**.**
*The absolute variability in healthcare demand increases with scale, whereas the relative variability decreases with scale. As a consequence, the buffer capacity required to handle demand variability decreases with scale, but the possible reduction becomes smaller as the scale increases.*

**Principle** **2**(Travel time)**.**
*The travel time for an efficient routing strategy is roughly proportional to the square root of the service area and number of clients. Moreover, the travel time is not subject to economies of scale.*

**Principle** **3**(Effective capacity)**.**
*Small teams are more prone to lower levels of effective capacity than large teams as a result of variability in leave of absence and sick leave, whereas the differences between larger teams become smaller.*

**Principle** **4**(Team composition)**.**
*Small teams must deploy above-average numbers of high-level care workers to sufficiently cope with the variability in demand. As the scale increases, the amount of capacity required will move closer to the average workload for each qualification level.*

**Principle** **5**(Contract type)**.**
*There is a restriction on how many contracts can be full-time, if split shifts need to be prevented. The fraction of full time contracts can be increased by augmenting the number of client-related care activities during the afternoon.*

**Principle** **6**(Communication and management)**.**
*The complexity of managing a team increases rapidly with team size due to the number of possible interactions between team members. The complexity can be mitigated by splitting the team into smaller flexible sub-teams coordinated by a central managing post.*

#### 3.2.1. Modeling Care Demand and Required Capacity

For Principle 1, we consider the amount of capacity that is required to keep home care accessible, i.e., avoid excessively long waiting lists. In particular, we determine the expectation and variability in the amount of care work that is offered. We use this to provide a rule of thumb for the required capacity that is provided by a rich literature on square-root staffing principles. Note that this principle relates to the direct care time in Figure 1 and the volume of care in Figure 2.

#### Model

First, we determine the demand for home care in terms of the required number of care hours per week if all demand can be met. Essentially, we interpret the demand for care as a discrete-time infinite-server queue, in which each server represents a single care hour per week. In particular, in line with Section 3.1, the three ingredients generating demand for care are (i) $A_{s}$, the number of new cases in week *s*; (ii) $S_{i}$, the length of stay (LoS) of case *i* (in weeks); and (iii) $B_{i}$, the demand for care per week of case *i* (in hours per week). We assume that the number of new clients, the LoS, and the case demand for care per week are all i.i.d. and mutually independent (see Remark 2 in case $S_{i}$ and $B_{i}$ are dependent). Moreover, we denote by $m_{a}$, $m_{s}$, and $m_{g}$ their respective means, and by $\sigma_{a}^{2}$, $\sigma_{s}^{2}$, and $\sigma_{g}^{2}$ their respective variances.

Next, we determine the mean and variance of the demand. Interestingly, the variance of the demand in stationarity can be expressed in terms of the so-called Gini coefficient (see also [36]). This coefficient is related to the Lorenz curve, which is used in economics to represent the inequality in the distribution of wealth or income among the citizens of a country. Here, we use it for the inequality in the LoS *S* among cases. The Gini coefficient is defined as the area under the Lorenz curve. In particular, the Gini coefficient [37] is, in this case for a discrete random variable *S*,
$$G_{s}=1-\frac{1}{ES}\sum_{k=0}^{\infty }P{\left(S>k\right)}^{2}.$$

For short- to medium-term planning of capacity of several weeks ahead, it is of interest to consider the time-dependent demand $N_{t}$. Let ${\hat{A}}_{0}$ be the number of patients currently present, i.e., at time 0, and let $S_{i}^{r}$ be their remaining LoS and ${\hat{B}}_{i}$ their case demand.

**Lemma** **1.**
*The mean and variance of the number of care hours t weeks from now is given by*

(1)
$$\begin{array}{cc} E[N_{t}] & =N_{0}P\left(S^{r}\geq t\right)+m_{a}m_{g}\sum_{s=0}^{t-1}S\left(t-s\right) \end{array}$$


(2)
$$\begin{array}{cc} V\mathrm{ar}(N_{t}) & =S^{r}\left(t\right)\left(1-S^{r}\left(t\right)\right)\sum_{i=1}^{{\hat{A}}_{0}}{\hat{B}}_{i}^{2}+m_{a}\left(\sigma_{g}^{2}+m_{g}^{2}\right)\sum_{s=0}^{t-1}S\left(t-s\right)\\ & \;\;\;\;+m_{g}^{2}\left(\sigma_{a}^{2}-m_{a}\right)\sum_{s=0}^{t-1}S{\left(t-s\right)}^{2} \end{array}$$

*with S(t)=P(S≥t) and $S^{r}\left(t\right)=P\left(S^{r}\geq t\right)$. In stationarity, the mean and variance of the number of care hours reduce to*

(3)
$$\begin{array}{cc} E[N] & =m_{a}m_{s}m_{g} \end{array}$$


(4)
$$\begin{array}{cc} V\mathrm{ar}(N) & =m_{a}m_{s}m_{g}\left[\frac{\sigma_{g}^{2}}{m_{g}}+m_{g}+m_{g}\left(1-G_{s}\right)\left(\frac{\sigma_{a}^{2}}{m_{a}}-1\right)\right] \end{array}$$



**Proof.** Consider the required demand in week *t*. We then have the following relation:
(5)$$N_{t}=\sum_{i=1}^{{\hat{A}}_{0}}1\left\{S_{i}^{r}\geq t\right\}{\hat{B}}_{i}+\sum_{s=0}^{t-1}\sum_{i=1}^{A_{s}}1\left\{S_{i}\geq t-s\right\}B_{i},$$
where $S_{i}$ and $B_{i}$ represent the LoS and weekly demand of the *i*th case arriving in that specific week. Observe that the first term represents demand from cases currently present, whereas the second term is due to cases that are yet to arrive. Using this relation, we may determine the first and second moment of the demand. More specifically, combining this relation with Wald’s equation, we obtain
(6)$$\begin{array}{cc} E[N_{t}] & =\sum_{i=1}^{{\hat{A}}_{0}}{\hat{B}}_{i}P\left(S^{r}\geq t\right)+\sum_{s=0}^{t-1}E\left[A_{s}\right]P\left(S\geq t-s\right)E\left[B\right]\;\\ \; & =N_{0}P\left(S^{r}\geq t\right)+m_{a}m_{g}\sum_{s=0}^{t-1}S\left(t-s\right). \end{array}$$
with $N_{0}$ the current demand for care and S(t)=P(S≥t) the survival probability or tail distribution of the LoS.Now, for the variance we distinguish again between cases currently present and newly arriving cases. Note that $1\{S_{i}^{r}\geq t-s\}$ corresponds to a Bernoulli random variable with probability $S^{r}\left(t\right)=P\left(S^{r}\geq t-s\right)$, from which we directly retrieve the variance.For the cases that are yet to arrive, we use that if the random variable *N* is independent of the random variables $X_{1},X_{2},\ldots$, then $V\mathrm{ar}\left(\sum_{k=1}^{N}X_{k}\right)=ENV\mathrm{ar}\left(X_{1}\right)+V\mathrm{ar}N{\left(EX_{1}\right)}^{2}$; see, e.g., ([38], Equation (A.10)). We will apply the above with $X_{i}=1\left\{S_{i}\geq t-s\right\}B_{i}$. Note that $1\{S_{i}\geq t-s\}$ corresponds to a Bernoulli random variable with probability S(t−s)=P(S≥t−s). Moreover, observe that
$$\begin{array}{ccc} V\mathrm{ar}\left(1\{S\geq t-s\}B\right) & = & V\mathrm{ar}\left(1\left\{S\geq t-s\right\}\right)\left(V\mathrm{ar}B+{\left(EB\right)}^{2}\right)+{\left(E1\{S\geq t-s\}\right)}^{2}V\mathrm{ar}B\\ & = & S\left(t-s\right)\left(1-S\left(t-s\right)\right)\left(\sigma_{g}^{2}+m_{g}^{2}\right)+S{\left(t-s\right)}^{2}\sigma_{g}^{2}. \end{array}$$Combining the above, we obtain
(7)$$\begin{array}{cc} V\mathrm{ar}(N_{t})\; & =\sum_{i=1}^{{\hat{A}}_{0}}{\hat{B}}_{i}^{2}P\left(S^{r}\geq t\right)\left(1-P\left(S^{r}\geq t\right)\right)\\ \; & \;\;\;\;+\sum_{s=0}^{t-1}EA_{s}\left[S\left(t-s\right)\left(1-S\left(t-s\right)\right)\left(\sigma_{g}^{2}+m_{g}^{2}\right)+S{\left(t-s\right)}^{2}\sigma_{g}^{2}\right]\\ \; & \;\;\;\;+V\mathrm{ar}\left(A_{s}\right)S{\left(t-s\right)}^{2}m_{g}^{2}\\ \; & =\sum_{i=1}^{{\hat{A}}_{0}}{\hat{B}}_{i}^{2}S^{r}\left(t\right)\left(1-S^{r}\left(t\right)\right)+m_{a}\left(\sigma_{g}^{2}+m_{g}^{2}\right)\sum_{s=0}^{t-1}S\left(t-s\right)\\ \; & \;\;\;\;+m_{g}^{2}\left(\sigma_{a}^{2}-m_{a}\right)\sum_{s=0}^{t-1}S{\left(t-s\right)}^{2}, \end{array}$$
where the second equality follows from some rewriting.For the stationary demand *N*, we let t→∞ in (Equation 6) and (Equation 7), yielding the result. □

**Remark** **1.**
*We note that the demand for home care is related to the number of customers in a discrete-time infinite-server queue with batch arrivals ($G^{X}/G/\infty$). The difference of such a queue with our setting is that we assume that every customer that arrives in the same batch has the same service time. In addition, we do not assume that a customer requires a server, that is, we allow for fractional values.*


**Remark** **2.**
*We note that it may be argued that $S_{i}$ and $B_{i}$ are dependent due to the type of care activity of case i. In that case, the demand per activity type can be analyzed first, yielding (Equation 3) and (Equation 4) for its mean and variance. Then, the total demand simply follows by aggregating over the activity types.*


The infinite-server queues provide some fundamental insight into how to choose the capacity in systems with a large but finite number of servers, through a rich literature on heavy-traffic approximations. These heavy-traffic approximations are typically in the Quality-and-Efficiency-driven (QED) regime. More specifically, the suggested heavy-traffic approximation for similar models (see, e.g., [39]) is
(8)$$N_{t}\approx N\left(E\left[N_{t}\right],V\mathrm{ar}\left(N_{t}\right)\right),$$
where $N(\mu ,\sigma^{2})$ is a random variable of a normal distribution with mean μ and variance $\sigma^{2}$.

In [30], the author focuses on a rough characterization of the required service capacity to achieve a desired grade of service γ, where the grade of service is related to the probability of delay. Using (Equation 8), it can be seen that the approximate required capacity t=0,1,… weeks from now is $s_{t}=\rho_{t}+\gamma \sqrt{\rho_{t}z_{t}}$, which is also often referred to as the square-root staffing formula (which is intimately related to the QED regime). Here, $\rho_{t}=E\left[N_{t}\right]$ is the expected demand in week *t*, and $z_{t}=V\mathrm{ar}\left(N_{t}\right)/E\left[N_{t}\right]$ is called the peakedness, or VMR, reflecting the variability in the aggregated demand process. Observe that with this choice of $s_{t}$, it holds that the probability that the demand exceeds capacity *s* equals $P\left(N_{t}\geq s_{t}\right)=1-\Phi \left(\gamma \right)$. We note that there is now a substantial body of literature on such heavy-traffic approximations with many servers; see, e.g., the recent survey [29] and references therein. Moreover, similar types of asymptotic results have been derived for infinite-server queues with batch arrivals, see [39,40].

For the first principle, that is, the required capacity that HHC organizations need, we rely on the heavy-traffic approximations of many server queues. In particular, assuming that HHC organizations operate in a QED regime combined with Lemma 1, we can specify Principle 1 as follows.

**Principle** **1**(Care demand)**.**
*For some grade of service γ (typically γ∈[0.5,2]), the required weekly capacity C is*
(9)$$C=\rho +\gamma \sqrt{z\rho },$$
*where $\rho =m_{a}m_{s}m_{g}$ is the average demand, and z is the peakedness given by*
(10)$$z=\frac{\sigma_{g}^{2}}{m_{g}}+m_{g}+m_{g}\left(1-G_{s}\right)\left(\frac{\sigma_{a}^{2}}{m_{a}}-1\right).$$
*Hence, the utilization of the capacity C is*

$$\frac{E[N]}{C}=\frac{1}{1+\frac{\gamma \sqrt{z}}{\sqrt{\rho }}},$$

*revealing economies of scale and diminishing returns.*


Here, the peakedness *z* represents the variability in demand that results from variability in the arrival process, LoS, and case demand per week. To be precise, Var(N)=zρ. Observe that the first term in (Equation 9) ensures that the capacity is sufficient to handle the load on average, whereas the second term represents the safety capacity required to cover the variability in demand (in particular, the standard deviation of *N* is $\sqrt{z\rho }$). Hence, the safety capacity only grows with the square root of the offered load, providing opportunities for economies of scale.

The principle as stated above is formulated for a stationary system, i.e., for the long term in case of the absence of structural changes. For the short-term, in the order of weeks, the care demand depends on the current situation. The principle can easily be adapted by using $E[N_{t}]$ and $z_{t}=V\mathrm{ar}\left(N_{t}\right)/E\left[N_{t}\right]$ instead of ρ and *z*, respectively.

**Remark** **3.***We note that our peakedness z is consistent with the G/G/∞ results. Assuming $B_{i}\equiv 1$, (Equation 10) reduces to $z=1+\left(1-G_{s}\right)\left(\frac{\sigma_{a}^{2}}{m_{a}}-1\right)$. Moreover, due to the relation between interarrival times and number counts, it holds that $c_{IA}^{2}=\sigma_{a}^{2}/m_{a}$, with $c_{IA}^{2}$ the squared coefficient of variation of the interarrival times. This corresponds to the classical result due to* [41]. *We refer to* [30,42] *for additional background and to* [36] *for the relation between the peakedness and the Gini coefficient.*

#### Application

In Figure 4, the utilization of capacity E[N]/C is illustrated as a function of the average weekly demand ρ in hours. This illustration is based on teams 5, 29, and 32, which were chosen based on their features depicted in Figure 2. In particular, team 5 is characterized by a relatively low volume of weekly demand (E[N]=132.30) and low variability (z=6.99). Team 29 exhibits a moderate volume of weekly demand (E[N]=231.59) but experiences substantial variability (z=18.49). Finally, team 32 has a high volume of weekly demand (E[N]=370.87) and a moderate level of variability (z=9.01). For the three lines in Figure 4, the peakedness is held constant whereas average weekly demand varies. For each team, the utilization of capacity is marked for their current weekly demand and peakedness on the respective graph. As can be observed, the utilization of capacity increases with the average weekly demand, albeit at a decreasing rate, demonstrating economies of scale and the law of diminishing returns. This effect appears when comparing teams 5 and 32 since the utilization of capacity of team 32 is significantly higher than for team 5 due to a larger volume of weekly demand (whereas the peakedness is somewhat comparable). Although team 29 also has a larger volume of weekly demand than team 5, its utilization of capacity is lower due to the relatively high peakedness. This demonstrates that reducing the variability in weekly demand can also increase the utilization of capacity. Overall, we see that an average demand of at least a couple of hundred care hours per week seems desirable for an efficient HHC operation. This exceeds the current size of most teams.

#### 3.2.2. Modeling Travel Time

Regarding Principle 2, the travel time of care workers is considered to be part of the direct care time in Figure 1. In this subsection, we describe a rule of thumb for the amount of travel time during a day part.

#### Model

The approximation for the travel time depends on the number of clients with care activities *n*, the size of the service area *A*, and the number of care workers *M*. Essentially, the travel time is the result of determining a set of *M* routes that visit all *n* clients from a central location (office of the HHC organization). Without any further constraints, this corresponds to a Vehicle Routing Problem (VRP), with vehicles corresponding to care workers. Due to many practical constraints, such as qualification levels and time windows, there is now a large body of literature on the Home Healthcare Routing and Scheduling Problem (HHCRSP); see, e.g., [9,10]. As there are currently no approximations for the route length of the HHCRSP, we consider the length of the optimal route for the VRP. However, note that similar approximations remain valid for the Capacitated VRP [43] and some specific time window instances [44], such that the approximation seems reasonably robust.

For the case M=1, already in 1959, [45] showed that the optimal tour in the classical TSP asymptotically converges to $k_{l}\sqrt{An}$ for n→∞ and the constant $k_{l}$. By now, there are various approximations [46], where many of them are of the following type:(11)$$\mathrm{VRP}\approx k_{l}\sqrt{An}+k_{c}\bar{r}M,$$
where $k_{l}$ and $k_{c}$ are constants, and $\bar{r}$ is the average distance from clients to the central location. The coefficients reported in [46] vary from 0.44 to 0.59 for $k_{l}$ and values close to 2 for $k_{c}$. Here, the first term corresponds to the length of the route required for visiting every client, whereas the second term relates to traveling from and to the central location.

The approximation (Equation 11) provides some interesting insights. First, the type of region (urban, suburban, and rural) influences the route length through the size of the area *A*, as may be expected. Second, as the number of visited clients *n* increases, the route length only increases with the square root of the number of clients. Hence, the travel time per client decreases and the relative amount of traveling becomes smaller. Finally, we give two illustrative examples to support the organizational decisions related to the team size.

*Example: merging regions.* Suppose that there are *R* identical neighboring regions, each with *n* clients, service area *A*, and *M* care workers. If the *R* regions are merged, there are nR clients, the service area is of size AR, and there are MR care workers. If the distance to the central location is the same, then the new total route length is
$$\mathrm{VRP}\;\text{-}\;\mathrm{merged}\approx k_{l}\sqrt{AR\times nR}+k_{c}\bar{r}MR=R\left(k_{l}\sqrt{An}+k_{c}\bar{r}M\right)=R\times \mathrm{VRP}.$$

Hence, there is no efficiency gain in traveling when merging different regions. In fact, the distance to the central location may become larger, making it even worse.

*Example: individual routes.* Suppose that the *n* clients are randomly assigned to the *M* care workers. This may happen when clients are pre-assigned to care workers to provide continuity of care. As the clients of each care worker may be spread over the area, the traveling time of each care worker is now $k_{l}\sqrt{A\times n/M}+k_{c}\bar{r}$. Hence, the total route length is
$$\mathrm{VRP}\;\text{-}\;\mathrm{ind}\approx M\times \left(k_{l}\sqrt{A\times \frac{n}{M}}+k_{c}\bar{r}\right)=\sqrt{M}\times \mathrm{VRP}+\left(1-\sqrt{M}\right)k_{c}\bar{r}M.$$

Apart from traveling from and to the central location, the route length becomes $\sqrt{M}$ times as large. Thus, pre-assigning clients to care workers may come at the cost of a considerable increase in traveling.

In practice, there can be various complicating factors, such as time windows and different qualification levels of tasks. However, the above examples and approximation provides some fundamental insight into routing to customers in a spatial area.

**Principle** **2**(Travel time)**.**
*The travel time for an efficient routing strategy is roughly $\mathrm{VRP}\;\approx \;k_{l}\sqrt{An}+k_{c}\bar{r}M$ for constants $k_{l}$ between 0.44 and 0.59, and $k_{c}$ close to 2. Moreover, merging regions does not lead to a more efficient route.*

#### Application

The total distance travelled by the team of care workers per day is estimated for each team using approximation (Equation 11). The area *A* corresponds to Table 1; we used the source data to determine the number of client visits *n* per day as this does not follow (directly) from Table 1. The client count includes instances where the same client was visited multiple times during a single day. As the teams did not operate from a central location, the average distance from clients to the central location is set to 0 ($\bar{r}=0$). Moreover, we took $k_{l}=0.5$.

The average of the approximated travel distances per area type can be found in Table 3. The relative differences between area types seem consistent with what would be expected; the travel distances in rural areas are notably longer than in urban and suburban areas, although there are variations between teams. Overall, given the reasonably small numbers, the contribution of traveling on the direct care time seems to be modest.

#### 3.2.3. Modeling Effective Capacity

Principle 3 concerns the availability of care workers. As visualized in Figure 1, care workers can be unavailable due to holiday and leave time (12.5%) and sick leave (currently 8.5%). The effective capacity *P* is defined as the fraction of time care workers are available, either for administration work or providing care to clients. Typically, management aims for a target effective capacity, where the mean effective capacity is currently 79% (see Figure 1). However, even if the target is met over the course of a year, a temporal shortage of care workers may occur due to randomness.

#### Model

To obtain insight into the impact of the team size on effective capacity, we consider the following stylized model. Let *M* be the total number of scheduled care workers during a period *T*, and let *p* be the probability that the care worker is present. The period *T* may either represent a single day, where *M* care workers are scheduled and 1−p is the probability of unexpected illness, or *M* may be the number of care workers over a longer period (e.g., summer holidays), and 1−p represents the probability a care worker is on leave. The number of care workers present $\tilde{M}$ then follows a Binomial(M,p) distribution. Consequently, the properties of the effective capacity $P=\tilde{M}/M$ follow directly from this observation.

**Principle** **3**(Effective capacity)**.**
*With p the probability that a care worker is present, the mean and variance of the effective capacity P are*
$$EP=p,\;and\;V\mathrm{ar}\left(P\right)=\frac{p(1-p)}{M},$$
*whereas P(P≤l), for l∈[0,1], follows from (Equation 12). Hence, for larger team sizes M, there is less variability in the effective capacity.*

In fact, from the above it follows that the standard deviation of *P* is linear in $1/\sqrt{M}$, showing economies of scale and the law of diminishing returns. Let us consider the impact of the team size *M* in more detail. We use the following representation for the binomial distribution, which also holds for non-integer *M*. For l∈[0,1], the probability that the effective capacity is at most *l* equals
(12)$$P\left(P\leq l\right)=P\left(\tilde{M}\leq lM\right)=\frac{B(lM+1,(1-l)M,p)}{B(lM+1,(1-l)M)},$$
with
$$B\left(x,y,p\right)=\int_{p}^{1}t^{x-1}{\left(1-t\right)}^{y-1}dt$$
the incomplete Beta function and B(x,y)=B(x,y,0).

**Remark** **4.**
*In practice, there may be variability in the number of working hours in period T of a care worker. Denote by $w_{i}$ the number of working hours of care worker i. Then, $P=\sum_{i=1}^{M}w_{i}{\tilde{M}}_{i}/C$ with $C=\sum_{i=1}^{M}w_{i}$, where ${\tilde{M}}_{i}$ is a Bernoulli random variable with probability p. Thus, we have*

$$EP=p,\;and\;V\mathrm{ar}\left(P\right)=p\left(1-p\right)\frac{\sum_{i=1}^{M}w_{i}^{2}}{C^{2}}.$$


*If $w_{i}=O\left(1\right)$ as C→∞, then we still have that $\sigma_{P}=O\left(1\right)/\sqrt{C}$ for C→∞, with $\sigma_{P}$ representing the standard deviation of P.*


#### Application

Figure 5 visually represents the probability P(P≤l) of having an effective capacity of at most *l*. In this case, we consider three thresholds: 50%, 60%, and 70%. The target effective capacity *p* is set at 79%. As an example, the two points in Figure 5 display the probability P(P≤0.6) for teams of size M=7 and size M=19. Those team sizes are based on a relative small team (team 5) and a larger team (team 32); the number of care workers per week is estimated by the ratio of the average weekly demand (Figure A1) and the average direct care time per week of a single care worker (estimated by our partner organization at 20 h per week). The figure shows that the probability of an effective capacity below 60% is around 0.21 for the small team, whereas this probability is only 0.05 for the larger team.

#### 3.2.4. Modeling Required Capacity per Qualification Level

Principle 4 concerns the team composition and relates to the direct care time of Figure 1 again. In Section 3.2.1, we established that the distribution of the weekly demand *N* can be approximated by a normal distribution. Under this normality assumption, we determine the capacity per qualification level (QL) required to cover demand, taking the hierarchy in qualification levels into account.

#### Model

Let Q={1,2,⋯,K}, K∈N be the set of qualification levels. We assume that there an ordering in QLs exists such that capacity of QL *k* can be deployed to cover the demand of all QLs j≤k as well. Denote $N_{k}$ as the required demand for QL k∈Q in a given week and assume that $N_{k}$ is normally distributed with mean $\mu_{k}$ and variance $\sigma_{k}^{2}$. Next, let $C_{k}\geq 0$ be the capacity of QL *k* deployed over the given week. The quantities $C_{k}$ can be viewed as (continuous) decision variables and should be chosen such that the capacities at least cover the average weekly demand per QL, whereas they also act as a buffer in case of demand fluctuations. Each unit of capacity of QL *k* comes at a cost $w_{k}\geq 0$, where we assume that $w_{j}\leq w_{k}$, if j≤k. Due to the minimum size of contracts, if capacity of QL *k* is used (i.e., $C_{k}>0$), then the corresponding capacity should at least be *ℓ* units.

Now, the (optimization) problem to determine the capacity per QL can be formally written as
(13)$$\begin{array}{cc} \min & \sum_{k\in Q}w_{k}C_{k} \end{array}$$
(14)$$\begin{array}{ccc} s.t.\; & C_{k}+e_{k}\geq \mu_{k}+\eta_{\alpha }\sigma_{k}, & \forall k\in Q, \end{array}$$
(15)$$\begin{array}{ccc} \; & C_{k}\in \left\{0\right\}\cup \left[\ell ,\infty \right), & \forall k\in Q, \end{array}$$
where $e_{K}=0$ and $e_{k}:=E{\left(\sum_{j>k}C_{j}-N_{j}-e_{j}\right)}^{+}$, for k<K, is the excess capacity of QLs k,…,K. Furthermore, $\eta_{\alpha }\geq 0$ in (Equation 14) is chosen such that $\Phi (\eta_{\alpha })=1-\alpha$, with α∈(0,1/2) to bound the probability of insufficient capacity. In particular, if (Equation 14) holds, then
$$P\left(N_{k}>C_{k}+e_{k}\right)=1-\Phi \left(\frac{C_{k}+e_{k}-\mu_{k}}{\sigma_{k}}\right)\leq \alpha .$$

For the expected excess over *C* of a normally distributed random variable *Y* with mean μ and variance $\sigma^{2}$, we have
(16)$$E{\left(C-Y\right)}^{+}=\left(C-\mu \right)\Phi \left(\frac{C-\mu }{\sigma }\right)+\frac{\sigma }{\sqrt{2}}e^{-\frac{1}{2}{\left(\frac{C-\mu }{\sigma }\right)}^{2}}.$$

Observe that the optimization problem (Equation 13)–(Equation 15) has a simple closed-form solution in case ℓ=0. In that case, it is always beneficial to set the capacity of each QL at the lower bound implied by (Equation 14) since $w_{j}\leq w_{k}$, for j≤k. In particular, the optimal solution to the optimization problem is then
(17)$$C_{k}=\min\left\{0,\mu_{k}+\eta_{\alpha }\sigma_{k}-e_{k}\right\},$$
which can be easily obtained by (backwards) induction using (Equation 16), starting at the highest QL *K*. The solution in (Equation 17) yields insight into the distribution in capacity over the QLs. Specifically, it follows directly that $C_{K}=\mu_{K}+\eta_{\alpha }\sigma_{K}$, implying that the highest QL has sufficient safety capacity. This may not hold for each individual QL, as capacity of higher QLs may be utilized.

Now, consider the capacity for QL *k* as a fraction of the total capacity, i.e., $C_{k}/\sum_{j\in Q}C_{j}$. To obtain insight into the impact of team size, we scale the demand by increasing the number of new cases $m_{a}$ per week while keeping the case demand and LoS the same. In view of (Equation 3) and (Equation 4), the demand $N_{k}$ is thus normally distributed with mean $m_{a}\mu_{k}$ and variance $m_{a}\sigma_{k}^{2}$. When $m_{a}$ grows large, the optimal capacities are given by (Equation 17) as $C_{k}$ will be larger than *ℓ* (we exclude the trivial case in which $\mu_{k}=0$). Then, it clearly holds that $C_{K}=m_{a}\mu_{K}+\eta_{\alpha }\sigma_{K}\sqrt{m_{a}}$. Also, it may be verified by induction that the expected excess capacity $E{\left(C_{k}-N_{k}\right)}^{+}={\tilde{e}}_{k}\sqrt{m_{a}}$ for the constant ${\tilde{e}}_{k}\geq 0$ that can be iteratively determined. Hence, $C_{k}=m_{a}\mu_{k}+\sqrt{m_{a}}\left(\sigma_{k}-{\tilde{c}}_{k}\right)$ for the constant ${\tilde{c}}_{k}\geq 0$. This implies that $C_{k}\leq m_{a}\mu_{k}+\eta_{\alpha }\sigma_{k}\sqrt{m_{a}}$ for k=1,…,K−1, meaning that any QL k<K has relatively less overcapacity than the highest QL *K*. Moreover, $C_{k}/\sum_{j\in Q}C_{j}\to \mu_{k}/\sum_{j\in Q}\mu_{j}$ as $m_{a}\to \infty$, implying that the capacity ratios of the different QLs are equal to the demand ratios as the team size grows large.

**Principle** **4**(Team composition)**.**
*The optimal team composition to cover the demand of different qualification levels 1,…,K can be obtained by the optimization problem (Equation 13)–(Equation 15). The highest qualification level has relatively high overcapacity $C_{K}\geq \mu_{K}+\eta_{\alpha }\sigma_{K}$, whereas for larger teams the optimal capacity ratios converge to the corresponding demand ratios $C_{k}/\sum_{j\in Q}C_{j}\to \mu_{k}/\sum_{j\in Q}\mu_{j}$.*

We like to emphasize that the principle above only relates to the delivery of care. Activities such as supervision are more often invested at higher QLs, but this is not yet incorporated as it depends on agreements within the HHC organization.

#### Application

In Figure 6, an example of the behavior of $C_{k}/\sum_{j\in Q}C_{j}$ is illustrated as $m_{a}$ increases with $\eta_{\alpha }=2$. The example is based on team 5 since it has a relatively low volume of average weekly demand and a relatively well-balanced ratio of QLs, as depicted in Figure A1. The optimal capacity ratio for the current demand per QL of team 5 (i.e., $m_{a}=1$) is illustrated with a dashed vertical line. Moreover, the ratios in mean demand per QL, $\mu_{k}/\sum_{j\in Q}\mu_{j}$ are illustrated with corresponding dashed horizontal lines. To ensure that care workers can work at their own QL, we require that $C_{k}/\sum_{j\in Q}C_{j}$ is close to $\mu_{k}/\sum_{j\in Q}\mu_{j}$ for each QL. This implies that, for each QL, the solid lines should be close to the horizontal dashed lines in Figure 6. It can be observed that for smaller teams (lower values of $m_{a}$), there is a relatively large overcapacity for the highest QL due to a large relative variability in demand; see the capacity ratio (green solid line) for VP niveau 3 in Figure 6. Consequently, a large amount of the demand for the mid-tier QL (PV niveau 3, indicated in red in Figure 6) is covered by the excessive capacity of the highest QL. In this case, we see that the capacity ratio of the lowest QL (PV niveau 2+) is closest to the ratio in demand since the blue solid and dashed lines are relatively close to each other. For all QLs, the capacity ratios converge to the ratios in demand.

#### 3.2.5. Modeling Required Contract Type

Contract type affects all elements of Figure 1, but as far as Principle 5 is concerned, we focus on the demand pattern across the day of the direct care time, combined with the non-client related administration time. In particular, the relatively small fraction of work in the afternoon (see Figure 3) poses a challenge for offering large contracts.

#### Model

To quantify the impact of the demand pattern across the day on the mix of full-time and part-time contracts, we consider the following stylized example. We assume that *short* shifts take place during one day part (morning, afternoon, or evening) and are of equal length. A *long* shift covers a short shift and part of the afternoon work. In particular, we scale time such that the length of a short shift is the basic time unit. The length of a long shift equals a>1 times the length of a short shift. Let $b_{FT}\geq 5$ and $b_{PT}\in \left(0,5\right]$ denote the number of short shifts required for a full-time and part-time contract, respectively. We assume here that any care worker can at most structurally work 5 days per week. Moreover, denote by $f_{2}$ and $f_{0}$ the fraction of work that needs to be carried out during the afternoon (care activities during day part 2) or that can be scheduled at any moment (e.g., administration), respectively.

**Principle** **5**(Contract type)**.**
*All contracts can be full time if*
(18)$$f_{0}+f_{2}\geq \min\left\{\frac{b_{FT}-5}{b_{FT}-b_{PT}},1-\frac{1}{a}\right\}.$$
*If (Equation 18) does not hold, then the maximum fraction of full-time contracts ($p_{FT}$) to avoid split shifts, due to the large fraction of client-related care activities in the morning and evening, satisfies*

(19)
$$p_{FT}\leq \frac{\left(f_{0}+f_{2}\right)b_{PT}}{b_{FT}-5+\left(b_{PT}-b_{FT}\right)\left(f_{0}+f_{2}\right)}.$$



**‘Proof’ of Principle** **5.**Equation (Equation 19) follows by considering the amount of work that needs to be done during short shifts. Due to the structure of long shifts and the amount of work during the afternoon, the fraction of work during long shifts can be at most $\left(f_{2}+f_{0}\right)\times a/\left(a-1\right)$. We assume that $f_{2}+f_{0}<1-1/a$, as (Equation 18) holds otherwise and the result is trivial. Equivalently, the fraction of work during short shifts is at least $1-\left(f_{2}+f_{0}\right)a/\left(a-1\right)$. The total capacity is $M[b_{FT}p_{FT}+\left(1-p_{FT}\right)b_{PT}]$, expressed in terms of number of short shifts, where *M* is the total number of care workers. Thus, the total amount of work that needs to be carried out during shorts shifts is $M[b_{FT}p_{FT}+\left(1-p_{FT}\right)$$b_{PT}]\times \left(1-\left(f_{2}+f_{0}\right)a/\left(a-1\right)\right)$.Now, consider the maximum number of short shifts available as a result of $p_{FT}$. Consider a care worker with a full-time contract. To respect the contract hours, the number of short shifts *x* for a full-time contract should satisfy $x+\left(5-x\right)a=b_{FT}$; hence, for a care worker with a full-time contract, there are $\left(5a-b_{FT}\right)/\left(a-1\right)$ short shifts. Hence, the number of short shifts available is at most $M[p_{FT}\left(5a-b_{FT}\right)/\left(a-1\right)+\left(1-p_{FT}\right)b_{PT}]$. There should be a sufficient number of short shifts available to cover the amount of work. That is,
$$M\left[p_{FT}\frac{5a-b_{FT}}{a-1}+\left(1-p_{FT}\right)b_{PT}\right]\geq M\left[b_{FT}p_{FT}+\left(1-p_{FT}\right)b_{PT}\right]\times \left(1-\left(f_{2}+f_{0}\right)\frac{a}{a-1}\right).$$The inequality above can be rewritten as
$$p_{FT}\left(b_{FT}-5+\left(b_{PT}-b_{FT}\right)\left(f_{0}+f_{2}\right)\right)\leq \left(f_{0}+f_{2}\right)b_{PT}.$$In case $f_{0}+f_{2}\geq \left(b_{FT}-5\right)/\left(b_{FT}-b_{PT}\right)$, this equation holds for any $p_{FT}\in \left[0,1\right]$ yielding (Equation 18). Otherwise, solving for $p_{FT}$ yields (Equation 19). □

#### Application

In Figure 7, the right-hand side of the inequality in (Equation 19) is illustrated as a function of $f_{0}$ and $f_{2}$ for set values of a=1.5, $b_{FT}=8$, and $b_{PT}=5$. These values correspond to short shifts of 4 h, long shifts of 6 h, full time contracts of 32 h per week, and part time contracts of 20 h per week. In the current situation, the fraction of work during the afternoon is $f_{2}=7.3\%$ (as shown in Figure 3). The left blue dot in Figure 7 indicates that only about 15% of the contracts can be full time if other work cannot be carried out during the afternoon. If all administrative work can be carried out during the afternoon ($f_{0}=13\%$), then the second blue dot shows that the maximum fraction of full-time contracts a HHC provider can offer is approximately 50%. As an example, if the desired maximum fraction of full-time contracts is 80%, then this can be achieved by increasing the fraction of work during the afternoon or any moment ($f_{0}+f_{2}$) to approximately 32% (as indicated by the red star in Figure 7). This can potentially be achieved by shifting work from the (late) morning or (early) evening to the afternoon.

#### 3.2.6. Modeling Complexity of Team Size

The principles of the previous subsections indicate that the efficiency of a team (almost) invariably increases as the team size grows. However, it is also intuitively clear that larger teams are harder to manage. With this in mind, Principle 6 focuses on the number of interactions that occur within a team. For example, in [47] the author concludes that smaller teams make for better team work, mainly because information sharing between team members and coordinating activities among team members becomes more difficult as the team size grows. Although the discussion in [47] concerns project teams in a broader sense, the idea of considering interactions also applies to a home care context as care workers discuss the health status of their clients and coordinate their schedules.

#### Model

To illustrate the number of interactions in a team, we may represent a team of size *M* as a graph, where each node represents a team member and each edge corresponds to a line of communication between team members (i.e., interaction). Under the assumption that such a graph is complete (i.e., each team member is able to communicate with all other members within the team), there are
(20)$$\frac{M(M-1)}{2}$$
edges in total. We refer to Figure A4 for an illustration of a complete graph for M=5 and M=10. In terms of complexity, the number of interactions between a team of size *M* thus equals $O(M^{2})$. The blue line (for the complete team) in Figure 8 visualizes how the number of interactions increases as the team size *M* grows.

Both figures indicate that the difficulty of managing a team increases rapidly as the team size grows. However, it is difficult to determine an appropriate threshold that remains manageable based on the number of interactions as this will depend on the type of work and the realized and/or required number of interactions.

The number of interactions within a large team can be reduced by splitting up the team into smaller (flexible) sub-teams and centrally connecting the sub-teams via a mediator. In practice, the individual sub-teams can still function as one team. The sub-teams should be sufficiently flexible such that they can also assist other teams when necessary where the information flow is via the mediator.

To determine the implications of this strategy in terms of complexity, consider *k* individual sub-teams of sizes $M_{1},M_{2},\ldots ,M_{k}$, and let $M=M_{1}+\ldots +M_{k}$. As mentioned, all team members within an individual team interact with each other but do not interact with members of another team. Moreover, we assume that there is one mediator that is able to interact with all members in each team.

Again, the number of interactions within sub-team j∈{1,2,…,k} equals $M_{j}\left(M_{j}-1\right)/2$, whereas the number of interactions with the mediator is equal to *M*. Hence, in total there are
(21)$$M+\sum_{j=1}^{k}\frac{M_{j}\left(M_{j}-1\right)}{2}$$
interactions taking place. In case the sub-teams are of equal size, that is, $M_{j}=\frac{1}{k}M$, then Equation (Equation 21) simplifies to
(22)$$M+\frac{1}{2}M\left(\frac{1}{k}M-1\right)=\frac{1}{2}M+\frac{1}{2k}M^{2}=\frac{1}{2}M\left(1+\frac{1}{k}M\right).$$

Clearly, the complexity is still equal to $O(M^{2})$; however, the number of interactions is reduced compared to a single large team, cf. (Equation 20). Specifically, as *M* grows the number of interactions reduces by a factor $\frac{1}{k}$ in the limit when *k* sub-teams connected via a mediator are created instead of a single large team:$$\frac{\frac{1}{2}M\left(1+\frac{1}{k}M\right)}{\frac{1}{2}M\left(M-1\right)}\to \frac{1}{k}\;\mathrm{as}\;M\to \infty .$$

**Principle** **6**(Communication and management)**.**
*The complexity of the number of interactions for a team of size M is $O(M^{2})$. However, by splitting the team into k flexible sub-teams of equal size, managed by one mediator, the number of interactions can be reduced by a factor k as M grows large.*

#### Application

The number of interactions is illustrated in Figure 8 in case of a single team (complete team) and for split ups in 2, 3 and 4 sub-teams of equal sizes. In the figure, we highlighted the cases of teams consisting of M=7 and M=19 members in total, which are based on teams 5 and 32, respectively (see Section 3.2.3). The blue dots represent the case of a single team (representing the current situation), whereas the orange dots demonstrate the number of interactions in the (hypothetical) scenario where the teams are split up into two equally sized sub-teams. The benefit of splitting up teams into (two) sub-teams is obviously greater for large teams than for small teams.

## 4. Practice Based Scenarios

It is clear from Principles 1, 3, and 4 that the efficiency of a team increases with team size. Conversely, Principle 6 illustrates the difficulty in managing larger teams; note that team size has no direct implications for Principles 2 and 5. Moreover, Principles 1, 3, and 4 are subject to the law of diminishing returns; most efficiency improvements can be achieved by merging relatively small teams into larger ones.

In practice, the capacity of a team can typically be increased by merging one team with another (i.e., pooling all team members of both teams to cover their shared demand). Naturally, this only makes sense when the geographical territories of the teams are closely situated. To illustrate how the principles in Section 3.2 can be used in practice, we consider a selection of 9 out of the 55 teams of the HHC organization. The 9 teams are merged one by one into larger teams, whereupon we consider the effects under Principles 1, 3, 4, and 6 at each step of the merging process. The merging process is illustrated in Figure 9, showcasing the centroid of the geographical locations of the nine selected teams (each labeled with its team ID). The marker size represents the mean weekly demand relative to the other teams. Each arrow indicates the next step in the merging process: starting with team 5, we first merge teams 5 and 15; subsequently, we merge team 39 with the cluster of {5,15}, and so on. The characteristics of each team (including the mean weekly demand) can be found in Table 4.

The effects of Principles 1, 3, 4, and 6 are illustrated at each step in the merging process in Figure 10; starting from the left, each point indicates that another team is added to the cluster. Here, Figure 10a shows the utilization of capacity E[N]/C, Figure 10b the probability that effective capacity falls below 60%, and Figure 10d the number of interactions within the team. Moreover, Figure 10c shows the difference between the capacity and demand ratio for each QL *k*, $\left(C_{k}/\sum_{j\in Q}C_{j}\right)-\left(\mu_{k}/\sum_{j\in Q}\mu_{j}\right)$, which is ideally just above zero for every QL. From the figures, we observe that there is a considerable improvement in capacity utilization, the risk reduction in shortage in effective capacity, and the capacity deviations in the first few steps of the merging process. However, the improvements diminish significantly following subsequent merging steps. Roughly, most of the improvements have been achieved after four teams have been merged, i.e., teams 5, 15, 39, and 41, corresponding to a total aggregate of almost 800 care hours per week. Moreover, the number of interactions is already significant for the combination of the four teams. This can be partly mitigated by splitting the combined team {5,15,39,41} into two flexible sub-teams managed by a central post (yellow line in Figure 10d).

To be more specific, when comparing team 5 to the combined team {5,15,39,41}, we see in Figure 10 that there is a relative increase of 13% in capacity utilization; a relative decrease of 82% of overcapacity for PV niveau 2+ and 45% for VP niveau 3; and a relative decrease of 97% in probability that the effective capacity falls below a level of 60%. On the other hand, by merging all nine teams, the relative increase in capacity utilization only improves marginally by 3% (hence, ultimately giving a relative increase of 16%) compared to the situation with 4 teams merged. The improvements of merging 9 teams over the first 4 teams are 3% and 18% in terms of overcapacity of PV niveau 2+ and VP niveau 3, respectively, and a 2% reduction in risk of a low effective capacity. In line with earlier observations, we conclude that the majority of the benefits occur in the first few steps of the merging process. Finally, the combined team {5,15,39,41} has 441 possible interactions. Excluding team 41 would give a drop to 182 interactions (i.e., 68% reduction). Conversely, merging an additional team (team 38) results in 784 interactions (i.e., 78% increase). This shows that the impact on the number of interactions is severe.

The observation above intuitively indicates that merging teams 5, 15, 39, and 41 may produce an appropriate balance between efficiency and manageability. However, such decisions depend on the relative importance of the individual components. To formalize this idea, we consider the following weighted objective function that needs to be maximized:(23)$$\lambda_{1}\frac{E[N]}{C}+\lambda_{3}P\left(P>l\right)-\lambda_{4}\sum_{k\in Q}{\left(\frac{C_{k}}{\sum_{j\in Q}C_{j}}-\frac{\mu_{k}}{\sum_{j\in Q}\mu_{j}}\right)}^{+}-\lambda_{6}\frac{M(M-1)}{2}.$$

Here, the first term is the utilization of capacity based on Principle 1, the second term is the probability that the effective capacity exceeds level l∈[0,1] based on Principle 3, the third term is the overcapacity of each qualification level relative to the mean demand based on Principle 4, and the fourth term is the number of (potential) interactions based on Principle 6. The weights $\lambda_{i}\geq 0$ represent the relative importance of component *i*. For the number of interactions, the weight $\lambda_{6}$ also serves as a scaling factor (as the units of the first three terms are in %). Note that each component completely depends on either team size or mean demand, which are both a direct consequence of the merging process. Hence, it is possible to find the optimal level at which teams need to be merged by evaluating the objective function at each step of the merging process. This idea is illustrated in Figure 11, where we set $\lambda_{1}=\lambda_{3}=\lambda_{4}=1$ and use different values of $\lambda_{6}$ to signify the impact of prioritizing team manageability. Clearly, the decision depends on the weights $\lambda_{i}$, reflecting the trade-off of the decision maker’s policy. Nonetheless, Figure 11 indicates that the combination of only two or three teams might be ideal, corresponding to a total aggregate weekly demand of 350–500 care hours.

## 5. Discussion

The principles presented in Section 3.2 provide guidance to managers and policy makers when making decisions about team size and composition in the context of home healthcare. The principles reveal that efficiency improves with team size, albeit more prominently for smaller teams due to diminishing returns. Moreover, it is demonstrated that the complexity of managing and coordinating a team becomes increasingly more difficult as team size grows. An estimate for travel time is provided given the size and territory of a team, as well as an upper bound for the fraction of full time contracts, if split shifts are to be avoided.

In addition to the team size and composition, we provide some other interesting observations. First, we were able to quantify the variability in home-care demand and the corresponding capacity requirements per qualification level. Second, we provide a rough estimate of the total travel time as a function of client count, area size, and the number of available care workers. Somewhat surprisingly, travel times are hardly affected by the scale at which teams are organized, e.g., merging regions does in essence not lead to more efficient routes. Third, we touched upon a typical problem recognized by Dutch HHC organizations; due to the lack of care demand during the afternoon, part-time contracts are inevitable. We provide an upper bound for the number of full-time contracts based on the afternoon care demand and administration time.

Whereas the six principles provide valuable practical insights into team size and composition, it is crucial to place them in the right perspective. First of all, the principles are of a generic nature (which we consider as their strength), but their application depends on the specific context. Their implementation and corresponding effects will invariably be influenced by the context in which they are applied. Most notably, the impact of scale on manageability, cooperation, and the quality and efficiency of team work is difficult to quantify in general. Moreover, it is typically impossible to capture all of the details of the application within a model; after all, a model is a simplified representation of a real-life situation. See [48] for a plea of the application of deductive modeling. Hence, managers and policymakers should regard any results obtained from these models as estimations rather than absolute certainties.

As indicated above, we believe that there is great potential by increasing the scale at which HHC teams operate. At this point, we like to emphasize that multiple ways exist to organize healthcare on a larger scale, next to the ‘straightforward’ merging of teams. During the last decade, various strategies have been investigated to achieve economies of scale for hospital wards while mitigating their drawbacks; see, e.g., [49,50,51,52]. A common aspect is that each team does not necessarily need to operate at a large scale as long as some flexibility is organized such that teams cooperate when peaks in demand occur. We think that such a design might be a practical first step to improve efficiency in the context of HHC.

Besides creating practical value, our goal is to trigger the OR community to address tactical and strategic challenges that HHC organizations are facing. The presented data and principles provide a solid basis for further research in which the principles can be further explored and/or extended. For instance, one possible direction is to model the interplay between care demand and the available capacity in terms of a queueing model. Developing such a model is intricate as in practice the capacity will not be constant over time and the admission policy also plays a vital role (see e.g., [53]). Moreover, the team composition is fundamental to all scheduling and routing problems that differentiate between skill sets of care workers. Therefore, the process of determining the necessary capacity per qualification level as outlined in Principle 4 can potentially be improved by specifically customizing it to a direct application. Finally, team manageability is essential for establishing an ‘ideal’ team size as it acts as a natural counterbalance to the economies of scale implied by most other principles. Clearly, the increasing complexity of coordinating large teams should eventually lead to a decline in effectiveness. As a consequence, a thorough and comprehensive modeling and evaluation of team manageability from an OR perspective is necessary. It is worth highlighting that those subjects have received (almost) no attention within the existing OR literature, despite their significance.

## 6. Conclusions

In this contribution, six model-based principles (i.e., rules of thumb) are presented and illustrated using real-life demand data. These principles provide guidance to managers and policy makers when making decisions about team size and composition in the context of home healthcare. In particular, the principles involve insights in capacity planning (Principles 1 and 4), travel time (Principle 2), available effective capacity (Principle 3), contract types (Principle 5), and team manageability (Principle 6). The principles concerning capacity planning and effective capacity generally state that the efficiency of a team improves as team sizes increase (due to economies of scale). However, smaller teams benefit more from this effect than larger teams due to the law of diminishing returns. In contrast, larger teams also imply an increase in the complexity of team coordination. The principle on team manageability shows that the complexity increases in a quadratic fashion with team size. Overall, it seems that an ideally sized team should serve (at least) approximately a few hundreds care hours per week.

## Figures and Tables

**Figure 1 healthcare-11-02935-f001:**
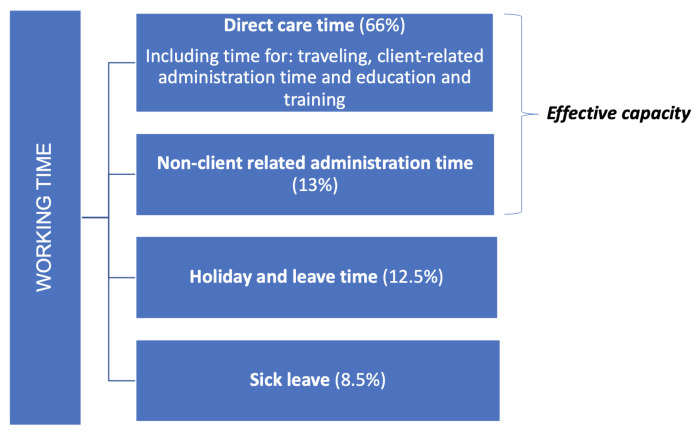
Distribution of working time across four categories.

**Figure 2 healthcare-11-02935-f002:**
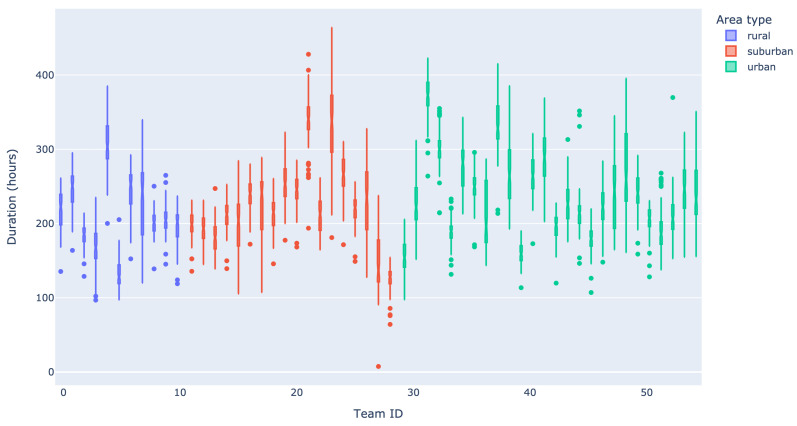
Boxplots of weekly demand for care services per team by area type; years = 2020 and 2021.

**Figure 3 healthcare-11-02935-f003:**
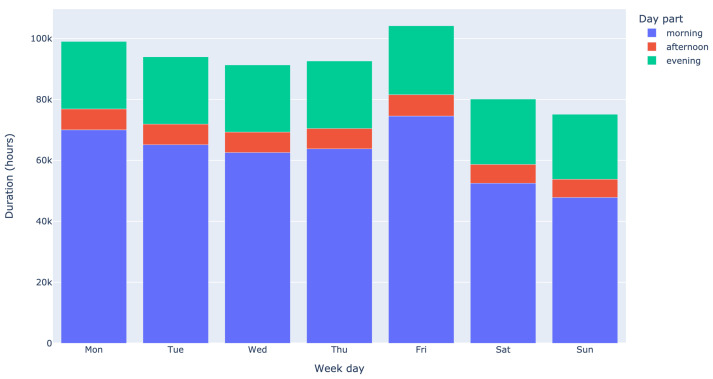
Total demand for care per week day by part of day; years = 2020 and 2021.

**Figure 4 healthcare-11-02935-f004:**
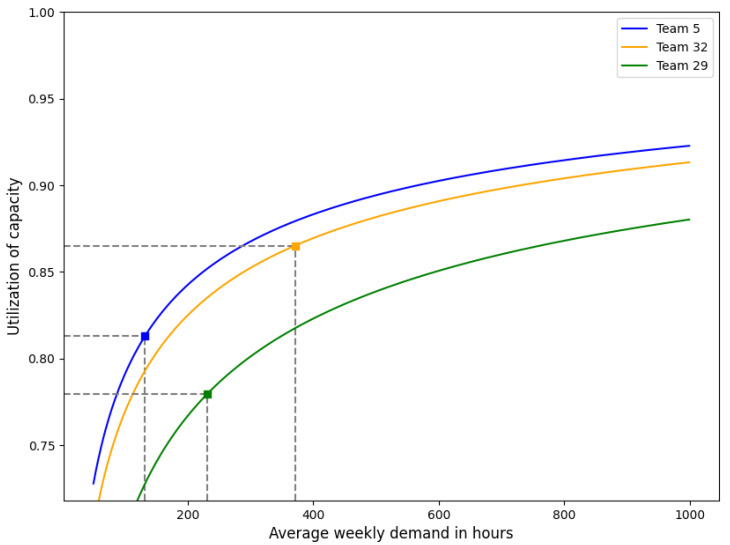
Utilization of capacity E[N]/C as a function of the average weekly demand ρ in hours, with γ=1 based on the peakedness of teams 5, 29, and 32.

**Figure 5 healthcare-11-02935-f005:**
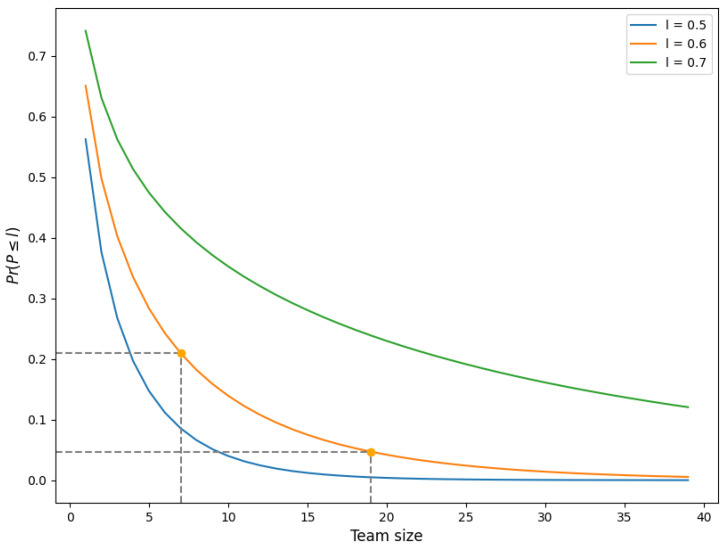
P(P≤l) as a function of the team size *M* for effective capacity levels *l* of 50%, 60%, and 70% (with p=0.79).

**Figure 6 healthcare-11-02935-f006:**
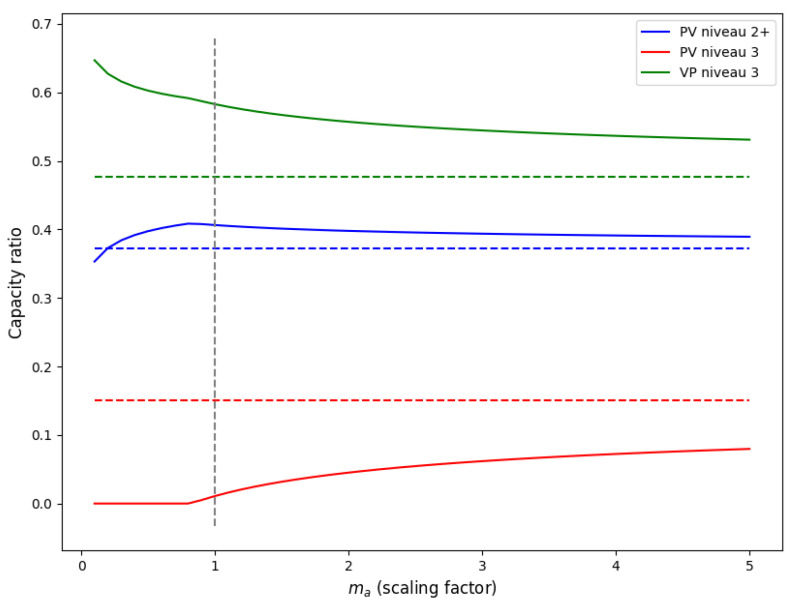
Ratio in capacity $C_{k}/\sum_{j\in Q}C_{j}$ as mean number of new clients $m_{a}$ increases; dashed horizontal lines represent ratios in demand $\mu_{k}/\sum_{j\in Q}\mu_{j}$. Dashed vertical line corresponds to current demand of team 5.

**Figure 7 healthcare-11-02935-f007:**
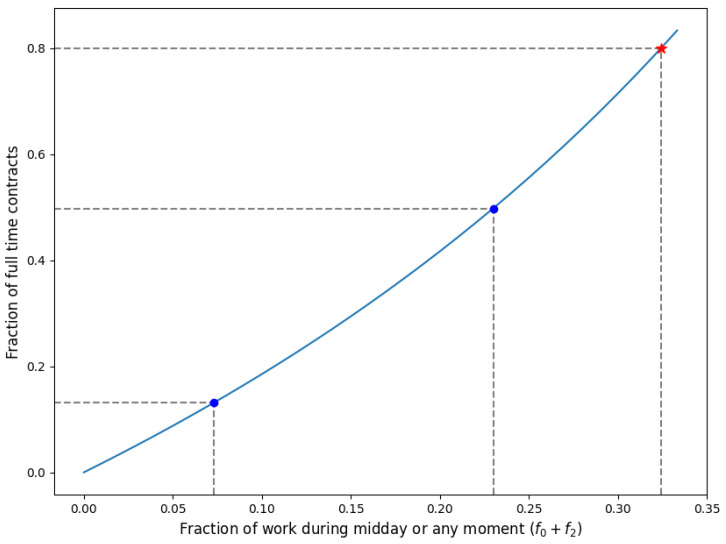
Maximum fraction of full-time contracts (upper bound of $p_{FT}$ in (Equation 19)) as a function of available work during the afternoon or any moment ($f_{0}+f_{2}$).

**Figure 8 healthcare-11-02935-f008:**
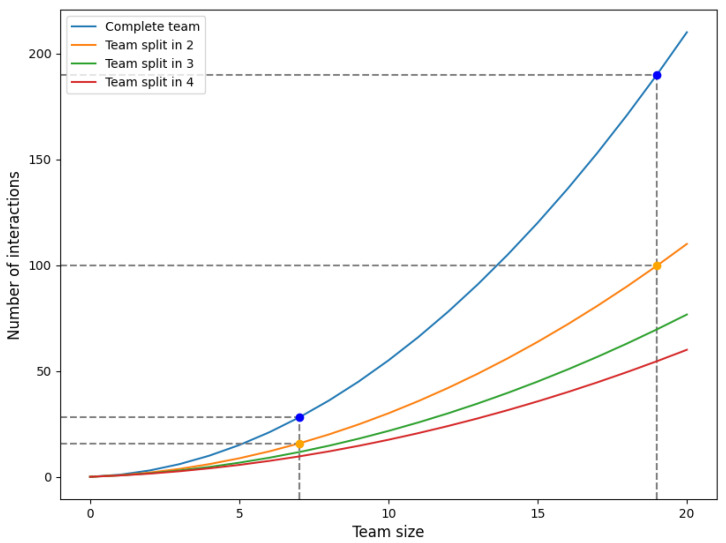
Number of interactions as a function of the team size (i.e., number of members) for a team where all members interact (complete team) and for teams that are split up and connected by a mediator.

**Figure 9 healthcare-11-02935-f009:**
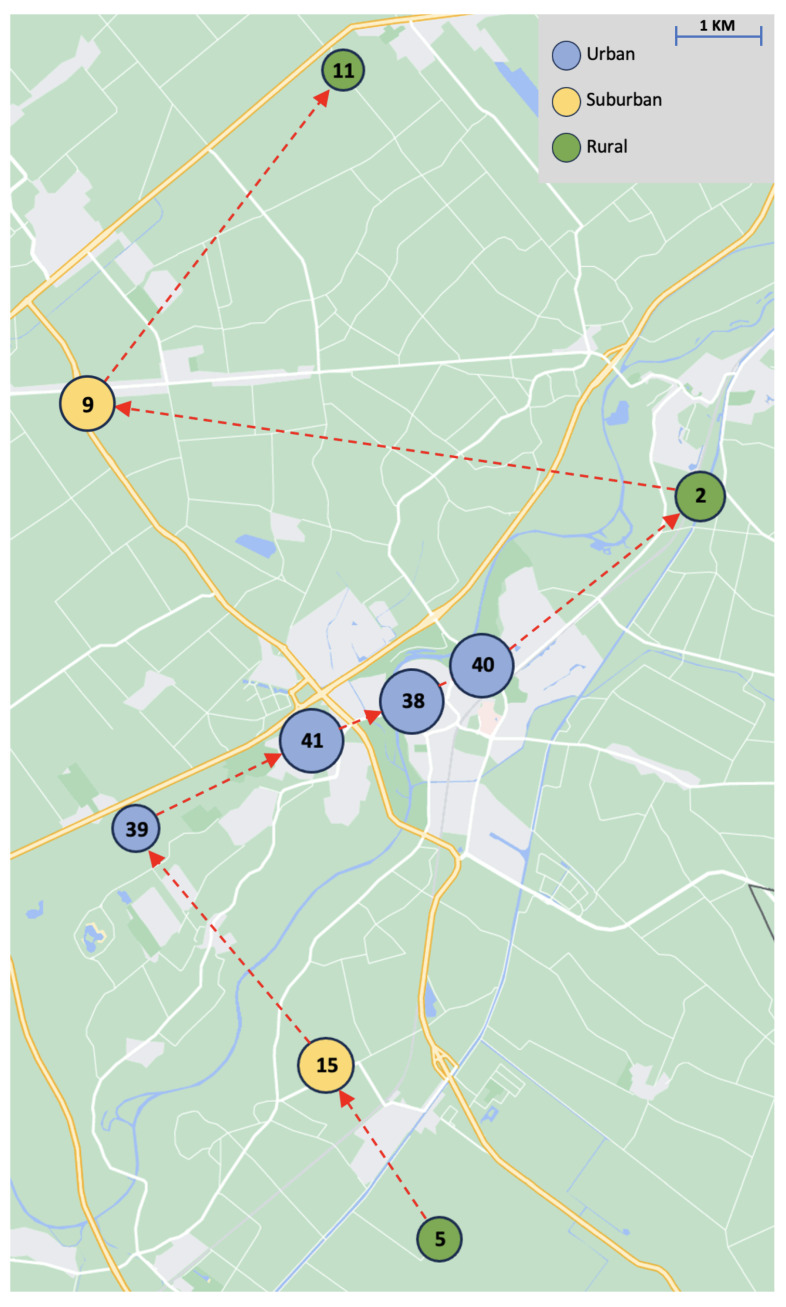
Geographical locations of the teams selected for the merging process.

**Figure 10 healthcare-11-02935-f010:**
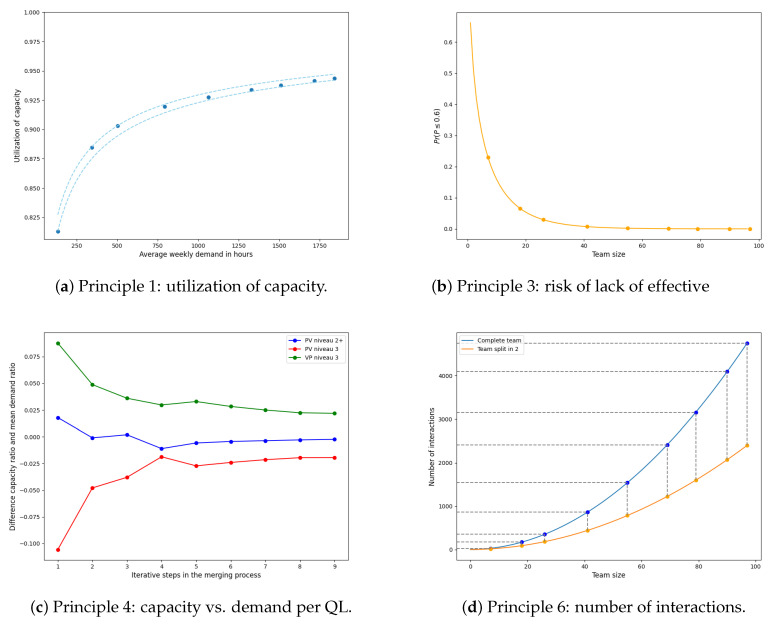
Illustration of Principles 1, 3, 4, and 6 at each step of the merging process; (**a**,**b**,**d**) correspond to Figure 4, Figure 5, and Figure 8, respectively, whereas (**c**) shows the difference between the ratios of optimal capacity $C_{k}/\sum_{j\in Q}C_{j}$ and mean demand $\mu_{k}/\sum_{j\in Q}\mu_{j}$ per QL.

**Figure 11 healthcare-11-02935-f011:**
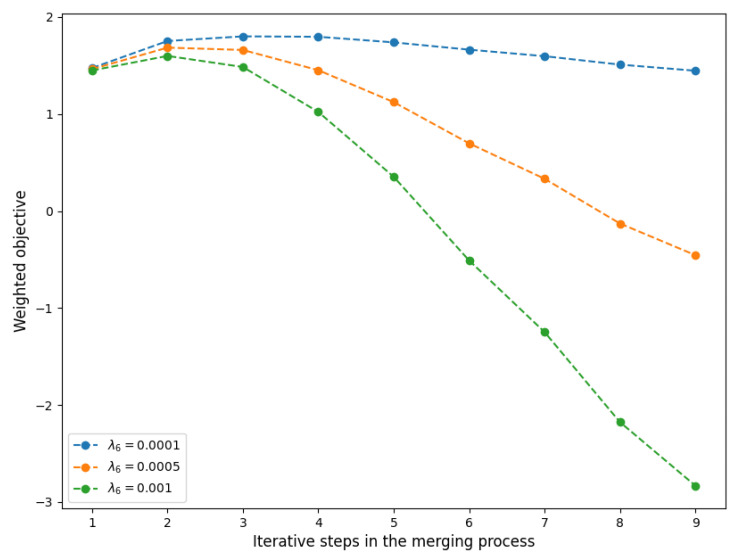
Objective function (Equation 23) at each step in the merging process for various weights of $\lambda_{6}$.

**Table 1 healthcare-11-02935-t001:** General information per area type for the years 2020, 2021; all numerical values (except team count) represent the mean, with standard deviation in brackets.

Area Type	Teams	Planned Care (h/wk)	Clients	Clients/${\mathrm{km}}^{2}$
Urban	26	238 (51)	221 (51.4)	71 (31.4)
Suburban	18	224 (54)	238 (61.3)	11 (5.4)
Rural	11	213 (46)	176 (53.6)	3 (1.1)

**Table 2 healthcare-11-02935-t002:** Overview of qualification levels.

Qualification Level	Description	Type of Tasks	Proportion of Total Planned Care
1	PV niveau 2+	Personal Care	67%
2	PV niveau 3	Personal Care	9%
3	VP niveau 3	Nursing	24%

**Table 3 healthcare-11-02935-t003:** Travel distance approximation per area type; all numerical values represent the mean, with standard deviation in brackets.

Area Type	Active Clients per Day	Total Distance (km) per Day	Travel per Client
Urban	83 (18.6)	10.0 (9.1)	0.12
Suburban	76 (24.7)	14.0 (7.2)	0.18
Rural	69 (16.4)	26.5 (6.0)	0.39

**Table 4 healthcare-11-02935-t004:** Characteristics of the teams selected for merging; see Section 3.2.1 for notation.

Team ID	Mean Demand	$m_{a}$	$\sigma_{a}^{2}/m_{a}$	$m_{g}$	$\sigma_{g}^{2}/m_{g}$	$G_{s}$	$m_{s}$ (Implied)
2	183.46	1.99	1.10	3.61	3.18	0.79	25.50
5	132.30	1.58	1.30	3.00	3.72	0.74	27.93
9	206.06	3.69	1.12	3.24	3.13	0.78	17.25
11	125.41	2.43	1.64	3.06	1.97	0.79	16.85
15	211.13	3.06	1.28	2.78	2.15	0.77	24.81
38	268.71	3.64	1.78	3.82	3.23	0.81	19.32
39	159.07	3.49	1.68	2.68	2.51	0.78	17.03
40	265.53	3.04	1.19	3.60	3.50	0.75	24.29
41	290.75	5.33	0.90	3.31	3.44	0.82	16.49

## Data Availability

Data are not publicly available.

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
