# Peer review of "Team Size and Composition in Home Healthcare: Quantitative Insights and Six Model-Based Principles"

_healthcare, 2023, doi:10.3390/healthcare11222935_

Round 1
Reviewer 1 Report
Comments and Suggestions for Authors
Home health care is an often-overlooked resource for our aging population. Maximizing the health care team so that all patients are served is an important consideration. Overall, I would say this is a very thorough analysis that could be duplicated with data sets from other home health care facilities. There are too many figures (15). I recommend carefully thinking about the audience (health care managers?) and how they would use the information to make decisions about their facilities and/or staff. The notation about minimum use being in the afternoon is very interesting and how shifts can be adjusted to have a break during this time or to use this time for documentation requirements, etc. (non-patient care tasks) can be recommended. The manuscript is well-written but the in-text citations such as "The study of [17] provides insights..." should be corrected to flow better.
Author Response
Dear Reviewer 1,
Thank you very much for reviewing our manuscript and the kind words. Your comments and suggestions were very helpful and allowed us to improve the manuscript. Please see below, in blue, for a point-by-point response to your comments and suggestions. All page numbers refer to the revised manuscript file with tracked changes. We hope that the revised version of our manuscript meets your expectations.
- There are too many figures (15). I recommend carefully thinking about the audience (health care managers?) and how they would use the information to make decisions about their facilities and/or staff.
Response: Thank you for pointing this out. We agree that there are many figures in the manuscript. We tried to solve this problem by looking carefully at the relevance of the figures (as suggested) and moving some figures to the (newly created) appendix.
- The notation about minimum use being in the afternoon is very interesting and how shifts can be adjusted to have a break during this time or to use this time for
documentation requirements, etc. (non-patient care tasks) can be recommended.
Response: Thank you for the compliment. Indeed, Principle 5 implicitly shows that the afternoon period is best utilized by adjusting shifts in such a way that administration and other non-patient care tasks are carried out in the afternoon.
- The manuscript is well-written but the in-text citations such as "The study of [17] provides insights..." should be corrected to flow better.
Response: In our opinion, this is a valuable comment. As suggested, throughout
Section 1 we have improved the flow (regarding references).
Reviewer 2 Report
Comments and Suggestions for Authors
Thank you for the opportunity to review this paper. It is an interesting article on an important topic and I enjoyed reading it. I offer the following points for you consideration:
* Please include more detail on the methods used to derive the principles in the abstract. I was a little confused when I started reading this as I wasn't sure what type of paper it is.
* Please include more detail on each study in table 1- just including reference numbers does not add much to the paper over just listing the number of studies in each domain and hierarchy level.
* It would be useful to comment on what these principles mean for the patient. E.g. what would be the impact of more part time workers to meet the constraints of principle 5?
I look forward to reading the next version of this paper!
Author Response
Dear Reviewer 2,
Thank you very much for taking the time to review the manuscript and your kind and encouraging words. Your comments and suggestions were very valuable and allowed us to improve the manuscript. Please see below, in blue, for a point-by-point response to your comments and suggestions. All page numbers refer to the revised manuscript file with tracked changes. We hope that the revised version of our manuscript meets your expectations.
- Please include more detail on the methods used to derive the principles in the abstract. I was a little confused when I started reading this as I wasn't sure what type of paper it is.
Response: Thank you for pointing this out. Based on this comment and a comment of one of the other reviewers, we revised the abstract.
- Please include more detail on each study in table 1- just including reference numbers does not add much to the paper over just listing the number of studies in each domain and hierarchy level.
Response: We agree with this comment. Based on this comment and comments from two of the other reviewers, we removed Table 1 from the revised version of the manuscript.
- It would be useful to comment on what these principles mean for the patient. E.g. what would be the impact of more part time workers to meet the constraints of principle 5?
Response: Thank you for this interesting suggestion. In general, you could say that the principles presented indirectly support better matching of care demand and available capacity. As such, applying the principles also provides added value from the perspective of the home care client. However, in our view, reflecting on added value from the client's perspective at a more detailed level is very difficult (if not impossible). Regarding principle 5: We consider the number of FT contracts as a function of the amount of work in the afternoon and administration time. Unfortunately, it is not easy rewrite the result to see the effect of having more part time workers. We hope that you and the editor can agree with us.
Reviewer 3 Report
Comments and Suggestions for Authors
Dear authors,
General aspects: the appropriate template proposed by the journal has not been used.
In relation to the general structure of the manuscript, the methodological design of the research is not appreciated; in addition, the IMRD structure recommendations are not followed. It is possible that I did not understand the study presented in the manuscript.
Abstract: it does not provide essential information about the study, such as the objective, methodology, results and main conclusions.
Keywords: Insufficients and not adjusted to MeSH descriptors.
References: In my opinion there are too many unjustified and very old references. References do not provide a link or doi to the primary source. The heading to the references section has not been indicated.
Introduction: the objective should be clearly placed at the end of the introduction and before the methodology section. An explanation of the process followed to present the manuscript is presented at the end of the introduction, but it does not follow a well-defined methodological approach.
Section 2. existing literature: does not adequately identify the process. There are doubts as to whether a review of the literature has been carried out or whether it just continues with the presentation of the background as part of the introduction section.
Another aim of the study is described on page 2, lines 129-129. This aspect creates inconsistency in the design.
Table 1 is not understandable; in my opinion it does not provide relevant information and is not self-explanatory.
Section 3, it seems unclear to me whether the authors are going to describe the analysis process (methodology) or describe results; however, they continue to report background information in their content. In section 3, it seems unclear to me whether the authors are going to describe the analysis process (methodology) or describe results; however, they continue to report background information in their content. I interprete that corresponds to results section, because authors reporting results from the contents of Tables 2 and 3, but up to this point the material and methods section has not been identified. I do not understand the subsections in which these results are organized as they have not been clearly explained.
The title of section 6 - Conclusions and discussion - is not described in a logical and coherent order. First the discussion and finally the conclusions should be presented.
Author Response
Dear Reviewer 3,
Thank you very much for your review and detailed comments about our manuscript. All your comments and suggestions have been carefully considered and the manuscript has been thoroughly revised accordingly. Please see below, in blue, for a point-by-point response to your comments and suggestions. All page numbers refer to the revised manuscript file with tracked changes. We hope that the revised version of our manuscript meets your expectations.
- Abstract: it does not provide essential information about the study, such as the objective, methodology, results and main conclusions.
Response: Thank you for this valuable comment. We revised the abstract accordingly.
- Keywords: Insufficients and not adjusted to MeSH descriptors.
Response: Thank you for pointing this out. As suggested, we revised the keywords to MeSH descriptors (https://meshb.nlm.nih.gov/). In the revised version of the manuscript the following keywords are used: Home care; Work force; Resource allocation; Efficiency.
- References: In my opinion there are too many unjustified and very old references. References do not provide a link or doi to the primary source. The heading to the references section has not been indicated.
Response: Thank you, this is an understandable comment. In the section ‘existing literature’ we looked at all papers that are relevant to our study. However, due to a lack of studies on the subject we decided to also include some older works. Furthermore, to underpin the principles we use fundamental work (where we preferred to reference the original work). Finally, we now added a DOI for each reference if available, and we added a header to the references section.
- Introduction: the objective should be clearly placed at the end of the introduction and before the methodology section. An explanation of the process followed to present the manuscript is presented at the end of the introduction, but it does not follow a welldefined methodological approach.
Response: Thank you for pointing this out. We revised the structure of the Introduction section by adding subsections. Regarding methodological approach, we added a Methods section.
- Section 2. existing literature: does not adequately identify the process. There are doubts as to whether a review of the literature has been carried out or whether it just continues with the presentation of the background as part of the introduction section.
Response: We believe the reviewer might have misunderstood our intention. The purpose of the literature overview, as presented in the Introduction section, is to get a “better grip” on the problem. As such, it complements the "background" subsection. We also briefly touched on the added value of this section in the Methods section (at step 1). We hope that you and the editor can agree with us.
- Another aim of the study is described on page 2, lines 129-129. This aspect creates inconsistency in the design.
Response: Thank you for pointing this out. However, we respectfully prefer not to change this. With this brief reflection we try to relate our contribution to the existing literature, without departing from the contribution as presented earlier in the introduction section. We hope that you reviewer and the editor can agree with us.
- Table 1 is not understandable; in my opinion it does not provide relevant information and is not self-explanatory.
Response: We agree with this comment. Based on this comment and comments from two of the other reviewers, we removed Table 1 from the revised version of the manuscript.
- Section 3, it seems unclear to me whether the authors are going to describe the analysis process (methodology) or describe results; however, they continue to report background information in their content. In section 3, it seems unclear to me whether the authors are going to describe the analysis process (methodology) or describe results; however, they continue to report background information in their content. I interprete that corresponds to results section, because authors reporting results from the contents of Tables 2 and 3, but up to this point the material and methods section has not been identified. I do not understand the subsections in which these results are organized as they have not been clearly explained.
Response: Thank you for this comment. Based on your suggestions we added a Methods and a Result section. In the newly added Methods section, we describe the research process in more detail following a literature-backed structure. Furthermore, we restructured the Introduction section by adding subsections.
- The title of section 6 - Conclusions and discussion - is not described in a logical and coherent order. First the discussion and finally the conclusions should be presented.
Response: Thank you for pointing this out. As suggested, “discussion” and “conclusions” are now two separate sections.
Reviewer 4 Report
Comments and Suggestions for Authors
First of all, I would like to appreciate the extensive research carried out by the authors on "Team size and composition in home healthcare: Quantitative insights and six model-based principles".
However, some minor corrections or suggestions from my side:
- What is the research design of your study?
- Is this research exempted from ethical approval?
- Line 34: That a vicious circle is lurking can be seen, for example, in the increasing absenteeism rates.
Is this a new sentence or continuation of previous sentence? If it is a new sentence, the sentence is not complete. If continuation, start with small "t".
- You mentioned reference number 8 as more recent. This study was published online on 22 Jun 2020, is this more recent?
- Line 107: From a healthcare capacity planning perspective determining team sizes can be considered a resource dimensioning issue.
Insert a comma between perspective and determining, to have ore clarity.
- Table 1 seems to be repetition of the details in the text. I suggest deleting the table.
- Give abbreviation of PV, VP niveau and explanation of 2+, 3.
- Figure 1 is repetition of text. I suggest deleting the text and keep the figure alone.
- Line 232: Provide abbreviation for QL (when you use for the first time). No need of abbreviation in line 814.
- Provide discussion followed by conclusion at the end.
-
Author Response
Dear Reviewer 4,
Thank you very much for taking the time to review our manuscript. We are grateful for your detailed comments and your insightful suggestions. It allowed us to improve the manuscript. Please see below, in blue, for a point-by-point response to your comments and suggestions. All page numbers refer to the revised manuscript file with tracked changes. We hope that the revised version of our manuscript meets your expectations.
- What is the research design of your study?
Response: Thank you for pointing this out. First of all, we added a Methods section in which we elaborate on the research design. Secondly, we also revised the abstract, in which we also included this aspect.
- Is this research exempted from ethical approval?
Response: Thank you for this valuable comment. For the purpose of this study, we did not use data that can be traced back to a person. We used "care cases" without using any personal data, and with regard to locations, we only used data at "area level" (in our case, area size and number of clients in that area). This is now also explicitly included in the form of an informed consent statement.
- Line 34: That a vicious circle is lurking can be seen, for example, in the increasing absenteeism rates. Is this a new sentence or continuation of previous sentence? If it is a new sentence, the sentence is not complete. If continuation, start with small "t".
Response: We have rewritten the sentence to make it more clear.
- You mentioned reference number 8 as more recent. This study was published online on 22 Jun 2020, is this more recent?
Response: We agree with this comment. We left the “the more recent” part out.
- Line 107: From a healthcare capacity planning perspective determining team sizes can be considered a resource dimensioning issue. Insert a comma between perspective and determining, to have ore clarity.
Response: We agree with this comment. We inserted a comma.
- Table 1 seems to be repetition of the details in the text. I suggest deleting the table.
Response: We agree with this comment. Based on this comment and comments from two of the other reviewers, we removed Table 1 from the revised version of the manuscript.
- Give abbreviation of PV, VP niveau and explanation of 2+, 3.
Response: Thank you for noticing that the skill level descriptions are not accompanied with an explanation. We now added an explanation of the descriptions after we introduce the concept of qualification levels.
- Figure 1 is repetition of text. I suggest deleting the text and keep the figure alone.
Response: We agree there is significant overlap between Figure 1 and the corresponding text. We removed the text and replaced it with a short passage that links the categories to their references.
- Line 232: Provide abbreviation for QL (when you use for the first time). No need of abbreviation in line 814.
Response: Thank you for noticing this. We made the changes accordingly.
- Provide discussion followed by conclusion at the end.
Response: Thank you for pointing this out. As suggested, “discussion” and “conclusions” are now two separate sections.
Round 2
Reviewer 3 Report
Comments and Suggestions for Authors
Dear authors,
Thanks for the improvements. The modifications made in this version have improved the comprehensibility of the study. In my opinion, the research subject matter addressed is highly complex and I understand that it is difficult to conform to a conventional structure for reporting the results.
In my opinion, the following changes should be made to improve the logical order in the contents:
1) Section 2 methods, should be described after the Contribution section (page 2, line 101). By making this modification, the current last paragraph of the Contribution section (page 2, lines 95-100) is redundant.
2) If you consider that the existing Literature section is not a result, it should be included in the background section, so it should also not be detailed in the steps described in the method (if you decide that this section (existing literature) needs to be described in the method, it should be moved after the description of the method, as noted in the comment above).
3) In the Contribution section, the aim should be moved to the end (at the end of the introduction/background), just before methods.
4) The way the manuscript has been organized requires the authors to continually explain the different sections and their structure. This aspect should be simplified by following the IMRYD structure. It is not appropriate that the result section starts with a first point (subsection 3.1) in which the background of the context should be explained (this aspect should be in the background).
5) Figure 1 should be shown after being cited in the text, not before.
Author Response
Dear Reviewer #3
Thank you very much for reviewing our manuscript again and the kind words. Please see below, in blue, for a point-by-point response to your comments. We hope that the revised version of our manuscript meets your expectations.
1) Section 2 methods, should be described after the Contribution section (page 2, line 101). By making this modification, the current last paragraph of the Contribution section (page 2, lines 95-100) is redundant.
Response: we adopted this in de revised version as proposed.
2) If you consider that the existing Literature section is not a result, it should be included in the background section, so it should also not be detailed in the steps described in the method (if you decide that this section (existing literature) needs to be described in the method, it should be moved after the description of the method, as noted in the comment above).
Response: we adopted this in de revised version as proposed. We went for the first option: “literature section is not a result”. As such, and as suggested, we therefore removed it from the Methods section.
3) In the Contribution section, the aim should be moved to the end (at the end of the introduction/background), just before methods.
Unfortunately, we do not fully understand this comment. Our sincere apologies for this. In the Contribution subsection, we present the purpose of our study and briefly explain it. With this, we hope that the Contribution subsection meets your expectations.
4) The way the manuscript has been organized requires the authors to continually explain the different sections and their structure. This aspect should be simplified by following the IMRYD structure. It is not appropriate that the result section starts with a first point (subsection 3.1) in which the background of the context should be explained (this aspect should be in the background).
Response: we adopted this in de revised version as proposed. For the sake of readability, we have divided the background subsection into "Background: Existing literature" and "Background: Practice".
5) Figure 1 should be shown after being cited in the text, not before.
Response: we adopted this in de revised version as proposed.